# Inverse design of chiral functional films by a robotic AI-guided system

Yifan Xie[1,5], Shuo Feng [1,5], Linxiao Deng [2], Aoran Cai[1], Liyu Gan[1], Zifan Jiang[1], Peng Yang[1], Guilin Ye[3], Zaiqing Liu[1], Li Wen[1], Qing Zhu [1], Wanjun Zhang[3], Zhanpeng Zhang[1], Jiahe Li[1], Zeyu Feng[1], Chutian Zhang[1], Wenjie Du[1], Lixin Xu[2], Jun Jiang [1] ✉, Xin Chen [4] ✉ & Gang Zou [1] ✉

Artificial chiral materials and nanostructures with strong and tuneable chir-optical activities, including sign, magnitude, and wavelength distribution, are useful owing to their potential applications in chiral sensing, enantioselective catalysis, and chiroptical devices. Thus, the inverse design and customized manufacturing of these materials is highly desirable. Here, we use an artificial intelligence (AI) guided robotic chemist to accurately predict chiroptical activities from the experimental absorption spectra and structure/process parameters, and generate chiral films with targeted chiroptical activities across the full visible spectrum. The robotic AI-chemist carries out the entire process, including chiral film construction, characterization, and testing. A machine learned reverse design model using spectrum embedded descriptors is developed to predict optimal structure/process parameters for any targeted chiroptical property. A series of chiral films with a dissymmetry factor as high as 1.9 ($g_{abs}$ ~ 1.9) are identified out of more than 100 million possible structures, and their feasible application in circular polarization-selective color filters for multiplex laser display and switchable circularly polarized (CP) luminescence is demonstrated. Our findings not only provide chiral films with the highest reported chiroptical activity, but also have great fundamental value for the inverse design of chiroptical materials.

Chiral materials/nanostructures with high and tuneable chiroptical performance are highly desired for applications in chiral sensing[1,2], enantioselective catalysis[3,4], 3D optical displays[5], quantum computing, and communication[6–8]. Much effort has been devoted to developing artificial chiral nanoparticles[9,10], assemblies[11–13], and metamaterials[14,15]. Their optical activities are often at THz or longer wavelengths, and the fabrication processes are usually sophisticated, time-consuming, and costly. Organic materials are flexible, inexpensive, and tuneable[16]. However, most organic materials only exhibit a weak chiroptical response ($g_{abs} < 0.1$), owing to the size mismatch between molecules

and visible photons[17]. Recently, layered nanomaterials that aligned with a twist angle have become a popular strategy; through the tuneable geometric handedness of these chiral stacked materials, their chirality can be manipulated in light-matter interactions[18], such as in two-dimensional (2D) van der Waals materials[19], twisted aligned graphene[20,21] and twisted metasurfaces[22,23]. Attempts to fabricate organic or polymeric chiral films via twisted-stacking have rarely been described to date[16,24–26]. One major obstacle is how to effectively explore the vast design space. In our chiral stacked system, numerous spectral and structural parameters significantly affect the resulting

[1]Key Laboratory of Precision and Intelligent Chemistry, School of Chemistry and Materials Science, University of Science and Technology of China, Hefei, Anhui, China. [2]State Key Laboratory of Particle Detection and Electronics, Department of Optics and Optical Engineering, University of Science and Technology of China, Hefei, Anhui, China. [3]Hefei JiShu Quantum Technology Co. Ltd., Hefei, China. [4]Suzhou Laboratory, Jiangsu, China. [5]These authors contributed equally: Yifan Xie, Shuo Feng. ✉e-mail: jiangj1@ustc.edu.cn; mail.xinchen@gmail.com; gangzou@ustc.edu.cn

chiroptical properties. These parameters include materials selection, thickness and strain of transparent films, absorption, thickness, strain, and greyscale of dyed films, as well as the twist angle between transparent film and dyed film. The combinations of these variables easily exceed ~$10^8$. Due to the richness of design choices, determining the optimal solution through conventional "trial & error" methods is very inefficient.

The benefit of combining an automated AI-Chemist with a computational brain has been demonstrated in the design and fabrication of films with optimal chiroptical performance[27]. AI-based methods have been widely used to accelerate the discovery of new materials, including photonic materials[28–31]. Most recently, a few automated synthetic platforms have been utilized for organic synthesis[32–35], flow chemistry[36], and so on[37–40]. The combination of high-throughput automated tools with machine learning methods[41,42] can significantly enhance the efficiency in material design, synthesis, and testing[43].

Herein, we designed an AI-guided automated platform that simultaneously achieves three important advancements. These advancements are: (i) accurate forwards prediction of chiroptical activities from the experimental absorption spectra and structure/process parameters, (ii) inverse generation of chiral films with giant and tuneable chiroptical activities at predesignated wavelengths across the full visible spectrum, and (iii) on-demand fabrication of films with user-specified chiroptical activities, including any target

circular dichroism (CD) spectral features such as wavelength and $g_{abs}$ values. The following real applications have been achieved using the AI chemist: (i) the generation of a set of circular polarization-selective color filters for multiplex laser display (NTSC color gamut value up to 140%) and (ii) the preparation of a series of fine-tuneable CP luminescent films ($g_{lum}$ value up to 1.9) using achiral perovskite quantum dots. The results not only exemplify the inverse design of chiroptical materials and devices with target functionalities, but also broaden the potential for discovering and optimizing new materials using the AI-Chemist.

## Results
### The overall workflow of the AI-Chemist
Figure 1 illustrates the overall workflow of the AI-Chemist with a computational brain to prepare chiral films with target chiroptical performance. Chiral films were prepared by stacking two different anisotropic layers in a twisted fashion, as schematically illustrated in Fig. 2a. The chiroptical response was described using the absorption dissymmetry factor $g_{abs}$ ($g_{abs} = (A_L - A_R)/((A_L + A_R)/2)$, in which $A_L$ and $A_R$ represent the absorption of left-handed and right-handed CP-light). The highest $g_{abs}$ values could almost reach the theoretical limit and the design principle of the heterostructured bilayer is described in the **Methods section** and Supplementary Fig. 1. This heterotwisted-stacking strategy endows not only an ultrastrong chiroptical

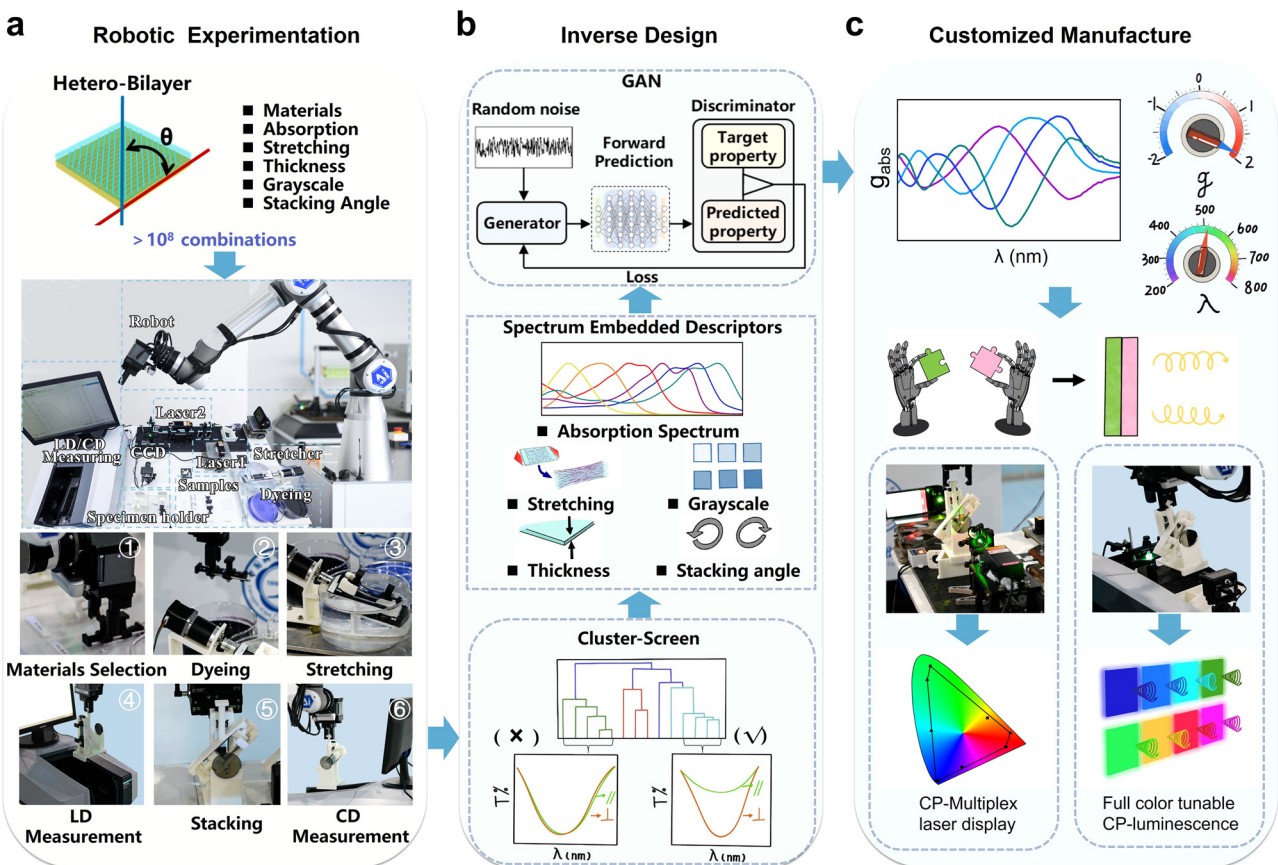

**Fig. 1 | Overall workflow to construct films with optimal chiroptical performance by the AI-Chemist. a** Illustration of the twist-stacking heterobilayer structure with variable parameters, resulting in over $10^8$ possible combinations, and the experimental process for creating these films using a robot platform involves the following steps: chiral film preparation (i: Materials Selection, ii: Dyeing, iii: Stretching and v: Stacking the films) and film characterization (iv: linear dichroism and vi: circular dichroism characterization); **b** The structural/process parameters are chosen via the following steps: (i) cluster analysis to prescreen the experimental parameters, (ii) quantitative structure-spectrum-activity relationship (QSSAR)

analysis using spectral embedded descriptors, and (iii) generative adversarial networks (GAN) for inverse design; **c** Inverse design-guided customized manufacture of chiroptical films with target functionality. Films with tuneable dissymmetry factor (*g*) values at any wavelength (λ) across the full visible spectrum can be inversely designed and fabricated on demand. Selective reflection and transmission of circularly polarized (CP) light with different frequencies from the heterostructure bilayer and full color tuneable circular polarized luminescence are demonstrated.

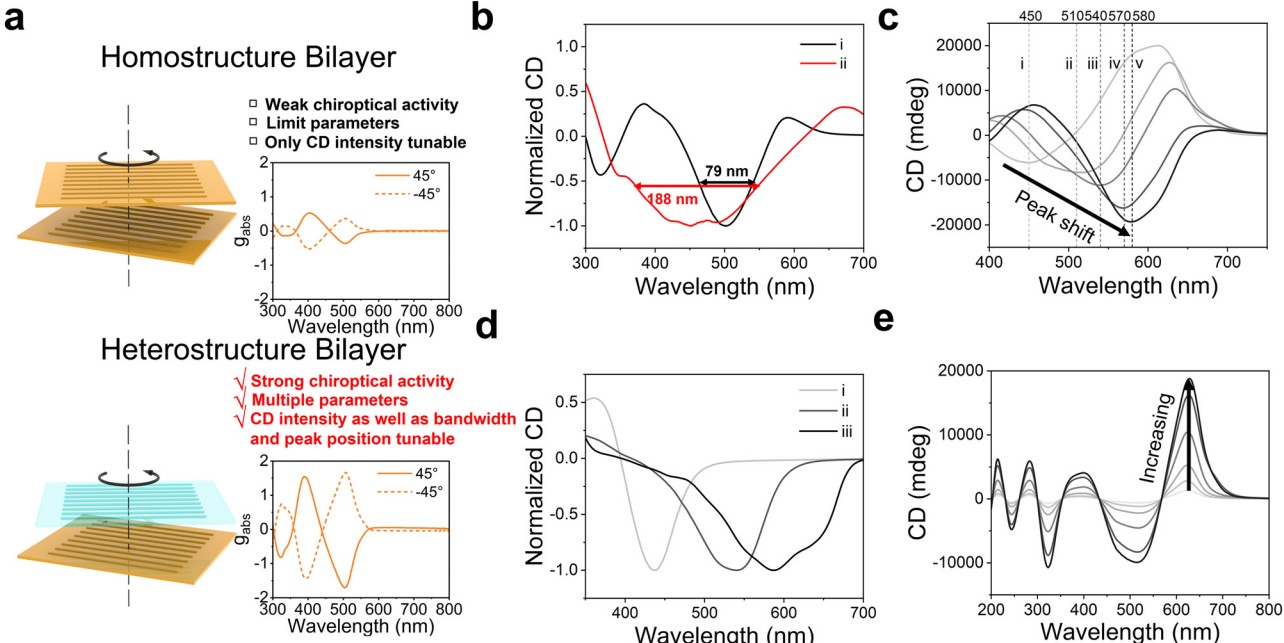

**Fig. 2 | Homo vs. heterostructured bilayer composite films with tuneable peak position, bandwidth, and circular dichroism (CD) intensity. a** Schematic illustration showing the structure and chiroptical performance of a homostructured bilayer vs. a heterostructured bilayer (top layer: polyvinyl alcohol (PVA), bottom layer: methyl orange embedded into PVA). Heterostructure bilayers often feature strong chiroptical activity and much greater design space (more multiple parameters to tune). **b** Bandwidth of CD spectra for heterostructure bilayer correlates with a dependence on thickness. (i: 80 μm, ii: 30 μm). **c** Peak position of CD spectra shifts (i: 450 nm, ii: 510 nm, iii: 540 nm, iv: 570 nm, v: 580 nm) to higher wavelength with greater strain (i: 80%, ii: 100%, iii: 133%, iv: 167%, v: 200%). **d** The peak of the CD spectra depends strongly on the wavelength of absorption (i: Sirius yellow, ii: Direct red 13, iii: Blue 6). **e** Intensity of CD spectra at about 628 nm correlates strongly with absorption intensity of the films at 618 nm (i: 0.10, ii: 0.14, iii: 0.18, iv: 0.26, v: 0.35, vi: 0.38 (absorbance (arb. units)).

response but also the freedom to manipulate the optical activity compared to the homostructured bilayer. There are several variable parameters, including matrix material selection, thickness and strain (or degree of stretching related to anisotropy) of transparent films, selective absorption of dye molecules (Supplementary Fig. 2), thickness, strain, and greyscale of dyed films, as well as the twist angle between transparent film and dyed film (Fig. 1a). In total more than $10^8$ possible combinations are estimated to achieve different chiroptical performances, including the sign, magnitude, center wavelength, bandwidth and the spectral profile (Fig. 2b–e). Optimizing the properties of the films is nontrivial, as these parameters are often correlated and there is no simple method to predict the chiroptical activity of the resulting films. In our previous work, we developed an all-round AI-Chemist that can conduct electrocatalyst and photocatalyst experiments[27]. Herein, an AI-chemist was recruited and adapted to solve this problem. The platform is capable of the following major functions: automatic experimentation using a mobile robot, machine learning-based inverse design, and customized manufacturing of chiroptical films with target functionality. The mobile robot platform is shown in Figs. 1a and 3a. It can move freely in the laboratory and locate its position using a combination of laser scanning coupled with visual feedback for fine positioning. Together with several automated workstations, the AI-chemist can carry out dexterous experiments that are comparable to those performed by human researchers, including (i) selecting materials; (ii) dispersing achiral dye molecules in a film; (iii) stretching a film to make it macroscopically anisotropic; (iv) characterizing linear dichroism (LD) properties; (v) stacking two anisotropic layers in a twisted fashion to construct a heterostructure film; and (vi) characterizing the chiroptical properties of the result film (for full details see **methods-film preparation and characterization**, Fig. 3b and Supplementary Movie 1). The machine learning-based inverse design was subsequently constructed and executed by the computational brain of AI-Chemist as shown in Fig. 1b. First, a cluster

analysis method was employed to screen the experimental parameters and reduce the parameter space. Therefore, a portion of the parameters were screened out and only five structure/process parameters were finalized. This resulted in a total number of combinations on the order of $10^5$ (see details in **methods-cluster-screening**, Supplementary Table 1). Second, the AI-chemist fabricated 1493 films by selecting from ~$10^5$ possible combinations of these parameters (see details in **methods-dataset**). The CD spectra of these films were subsequently measured. The spectral parameters (absorption of dye molecules) and four structural/process parameters (film construction) were combined into spectral embedded descriptors, which were used as the input for the forward model to establish the quantitative structure-spectrum-activity relationship (QSSAR). Last, due to the richness of design space, traditional forward design approaches based on trial-and-error are very inefficient. Machine learning-based QSSAR allows us to develop a reverse model for inverse design, which is adapted from the generative adversarial model[44], and consists of a generator and a discriminator. Furthermore, through the inverse design approach, target chiroptical properties can be customized, allowing the AI-chemist to identify optimal chiroptical films with target functionalities in the following practical applications—circular polarization-selective color filters for multiplex laser display and switchable CP luminescence (as demonstrated in Fig. 1c). The details are elaborated in the following sections.

**Prescreening experimental parameters**
First, prescreening was performed to reduce the parameter space for manufacturing the heterostructured films (Fig. 4). Two rounds of clustering-screening processes were executed based on transmittance ($T_0$) in which the polarization of the incident light was parallel to the anisotropy axis and linear dichroism (LD) of two monolayers, which were measured by the AI-chemist. The LD spectra were measured by a UV–vis spectrometer equipped with a polarizer. The LD peak was maximized when the optical axis of the dyed film was parallel to the

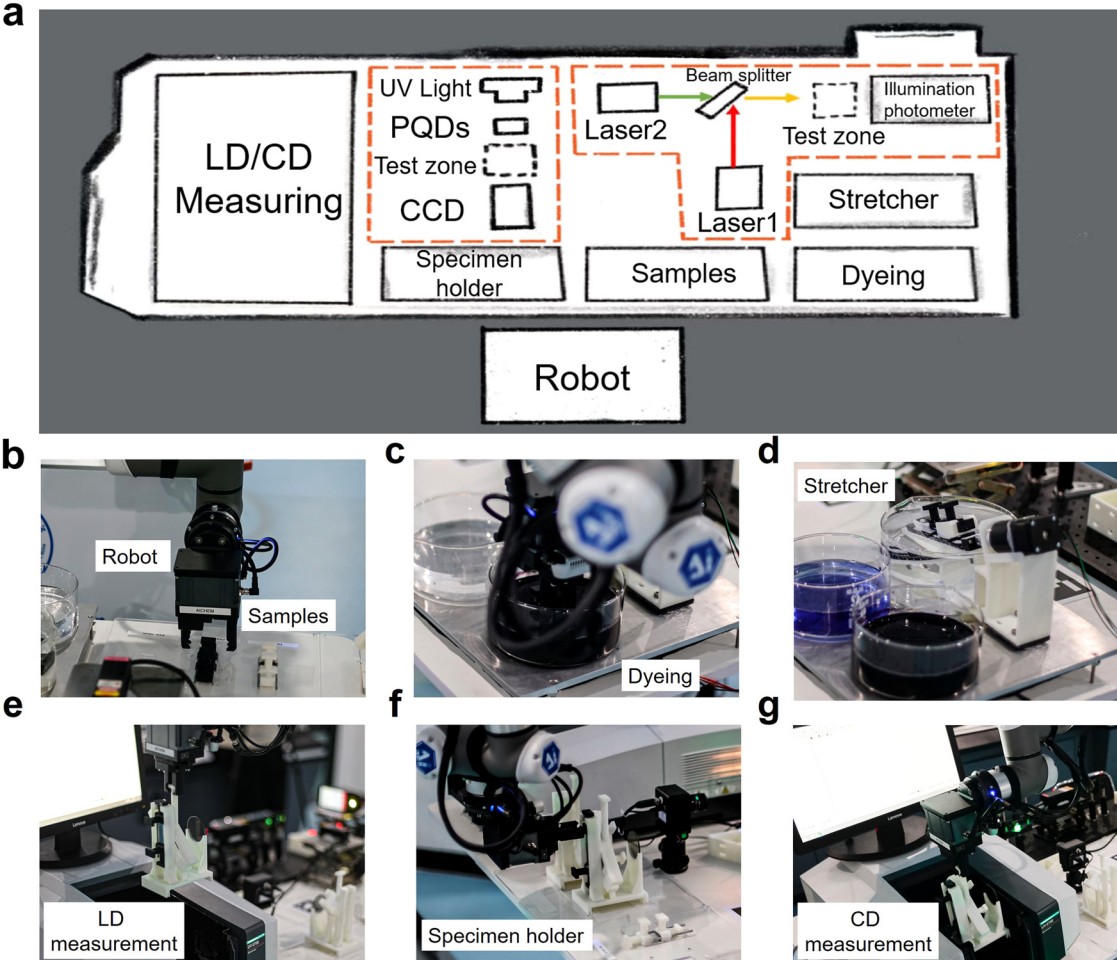

**Fig. 3 | Map of the work area and pictures of the AI-chemist executing key operations. a** Map of the work area used to construct, characterize, and test the chiral films, as well as perform the two real applications including selective reflection and transmission of circularly polarized (CP) light with different frequencies from a heterostructure bilayer and full color tuneable circular polarized luminescence associated with perovskite quantum dots (PQDs) which excited by ultraviolet (UV) light. **b** Selecting the materials; **c** dispersing achiral dye molecules in the film; **d** stretching the dyed film; **e** characterizing the linear dichroism (LD) properties of the layer; **f** constructing a heterostructure chiral film by stacking two different anisotropic layers in a twisted fashion; and **g** characterizing the dissymmetry factor ($g_{abs}$) of the twist-stacking structure.

polarization direction of incident light. After the optical axis of the film was rotated by 90°, an opposite peak with similar intensity appeared. Furthermore, a positive correlation was observed between the LD signal and strain increase (Supplementary Fig. 3). The parameter values with a high probability of producing a very low LD and very small $T_0$ were then removed. Only half of 20 dye molecule candidates passed this round of prescreening. Subsequently, the second round of prescreening further eliminated the films with a low LD or low $T_0$. PVA was selected as the material for the transparent films, and the thickness and strain of the dyed films were set to 80 µm and 400%, respectively. Thus, only the spectral parameters (the absorption of dye molecules) and four structure/process parameters (including thickness, strain of transparent films, greyscale of dyed films, and twist angle between transparent film and dyed film) were used as inputs to the model. The prescreening reduces the total number of possible combinations from $5 \times 10^8$ to $2 \times 10^5$ (see details in **methods-cluster-screening**, Supplementary Fig. 4–5, Supplementary Table 1), making it possible to build an accurate forward prediction model using data generated from a reasonable number of experiments.

**A forwards predicting model**

Next, a forward prediction model was established to predict the CD spectrum of a film. The AI-Chemist sampled 1493 combinations of the

parameters to construct films and measure their CD spectra with the robot. This dataset was used for both training and validation of a fully connected neural network (see **methods-dataset** and Supplementary Fig. 6–8 for details). The input vector of this network is spectral embedded descriptors, which include absorption strength at every 5 nm from 200 nm to 800 nm as well as the following structural/process parameters: twist angle, thickness, strain, and greyscale. The output is a vector representing the $g_{abs}$ value every 5 nm from 200 nm to 800 nm in the CD spectrum of the final film. This forward prediction network was trained and validated using 5-fold 80:20 splitting of the experimental data, as detailed in Supplementary Fig. 9 (see **methods-ML model**). The mean absolute error (MAE), root mean squared error (RMSE), and coefficient of determination ($R^2$) of the prediction model on the testing set are 0.04, 0.06, and 0.985, respectively. To further validate the accuracy of forward prediction, the model was checked against new experiments. Twenty sets of parameters were randomly generated, and the composite films were fabricated accordingly. The experimentally measured CD spectra were compared with the predicted spectra. The overall accuracy was excellent, and more details are shown in Supplementary Fig. 10. The forwards model also successfully predicts the varying trends of $g_{abs}$ with strain, thickness, and greyscale (Supplementary Fig. 11–13). The results confirm the accuracy and generality of the forward model.

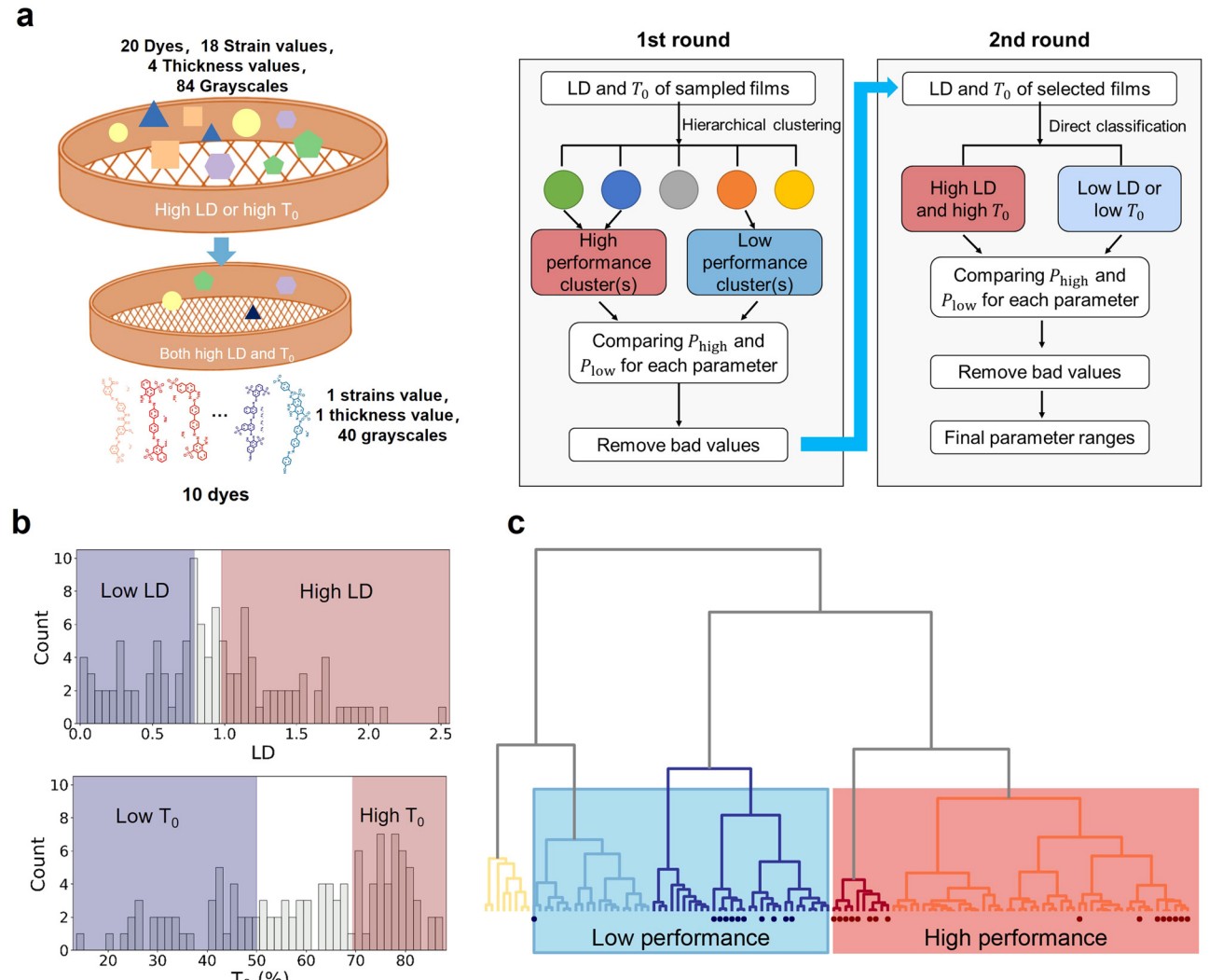

**Fig. 4 | Schematic illustration showing the clustering and prescreening of each film. a** Scheme of the two round clustering-screening process. In the first round, films were hierarchically clustered into several categories as follows: films with high linear dichroism (LD) and transmittance ($T_0$) in which the polarization of the incident light was parallel to the anisotropy were categorized as high performance, and those with both low LD and $T_0$ were categorized as low performance. The probability of each parameter value producing high-performance films ($P_{high}$) and low performance films ($P_{low}$) were calculated. If $P_{low}$ was significantly high (typically >60%), these values were deemed "bad choices" and removed. The second round was similar to the first round, except that the clustering process was replaced by direct classification, in which the low performance ones were defined as either small LD or low $T_0$. **b** Histogram of LD and $T_0$ for dyed films. The top 40% values are regarded as high performance (highlighted in red shading) and the bottom 40% values are low performance (highlighted in blue shading) in the first round. In the second round, the low thresholds are reduced by 40%. **c** After hierarchical clustering, dyed films form five clusters, where two are high performance clusters and two are low performance clusters.

## A generative model for the inverse design of chiroptical films

This quantitative structure-spectrum-activity relationship (QSSAR) obtained by machine learning was further exploited in a reverse model for inverse design, which consists of a generator and a discriminator, as outlined in Supplementary Fig. 14. The generator is a fully connected network with random noise as the input and a set of parameters, including the structural/process parameters and the type of dye molecule, as the output (see **methods-ML model** and Supplementary Fig. 15). This set of parameters is converted into spectrum embedded descriptors and used as one of the two inputs for the discriminator. The discriminator first recruits the machine learned QSSAR to form a prediction of the CD spectrum of a composite film through the parameters provided by the generator. Next, it calculates a "loss function", which is the difference between the predicted properties calculated from the predicted CD and the target properties and, is supplied by the user as the second input. Last, the "loss" is sent back to the generator as feedback, to update and improve the generator. The process is repeated recursively to reduce the "loss" to a preset threshold value. If successful, the outputs of the generator are considered the reverse design result, i.e., the experimental parameters that meet the target property. The target function is user defined and can take any form expressible from the CD spectral features, e.g., a certain $g_{abs}$ value at a certain wavelength. The target wavelength and $g_{abs}$ values were input into the computational brain, which can be executed with the reverse model and a set of parameters that approach the target were reported, as shown in Fig. 5a. Compared with the predicted spectra, the experimentally measured CD spectra exhibited excellent accuracy (Fig. 5b). We also used VHTS to obtain the results for several $g_{abs}$ values at several $\lambda$ as shown in Fig. 6 and Supplementary Fig. 16–20. There are typically multiple available answers for a given target. This is reasonable, as multiple films can have the same $g_{abs}$ value at a certain wavelength. Interestingly, the number of valid answers drops very quickly with the increase in the magnitude of target $g_{abs}$. This is expected since smaller $g_{abs}$ values (close to 0) indicate weak chiroptical activities, and

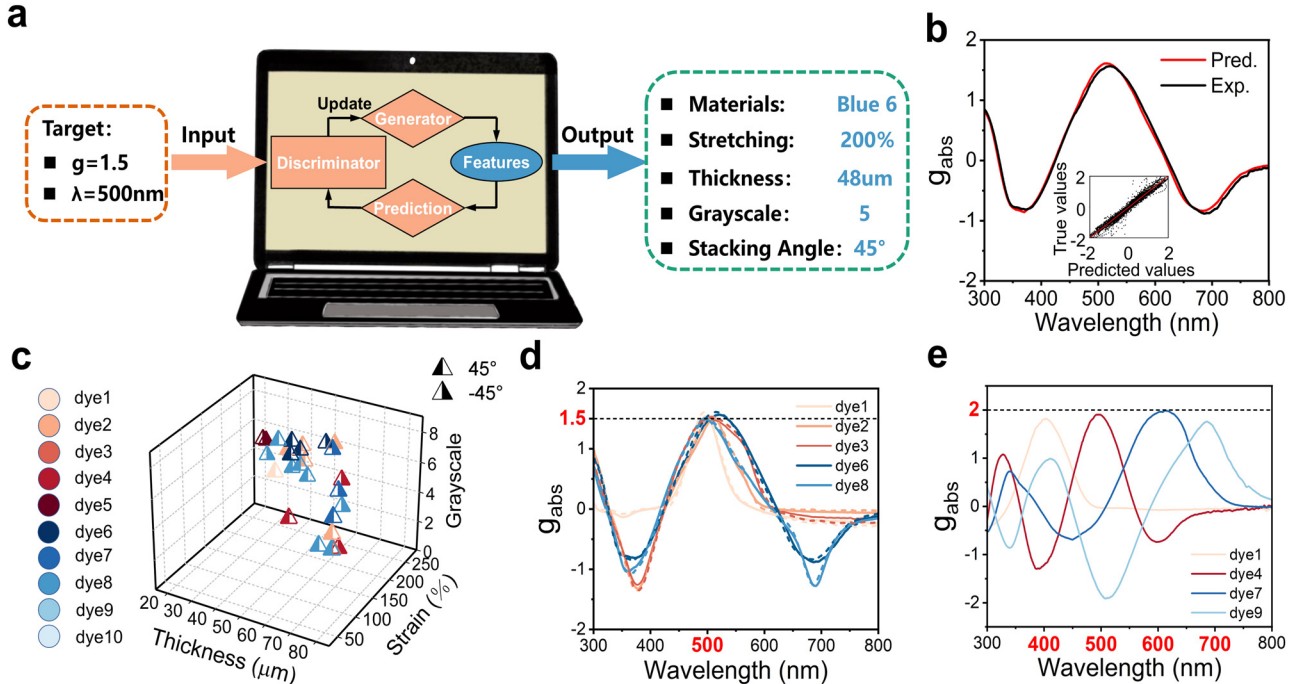

**Fig. 5 | Reverse model for inverse design of chiroptical membranes. a** Schematic diagram for the reverse design. **b** A typical example of a CD spectrum predicted with designed parameters vs. experimentally verified parameters. Inset: the error distribution of predicted vs. measured $g_{abs}$ values of 20 samples. The mean absolute error (MAE), root mean squared error (RMSE) and coefficient of determination ($R^2$) of the validation data are 0.09, 0.15, and 0.95, respectively. **c, d** Examples of experimental realization of inverse design for a target dissymmetry factor $g_{abs}$ = 1.5 at 500 nm. Films created with multiple sets of parameters can achieve this goal (**c** parameters generated by inverse design, **d** curves of experiment (solid lines) vs. prediction (dashed lines)). **e** Same as **c**, **d** but the target is set to $g_{abs}$ = 2, the theoretical limit, at 400 nm, 500 nm, 600 nm, and 700 nm. The reverse network provides the best available answer for each target.

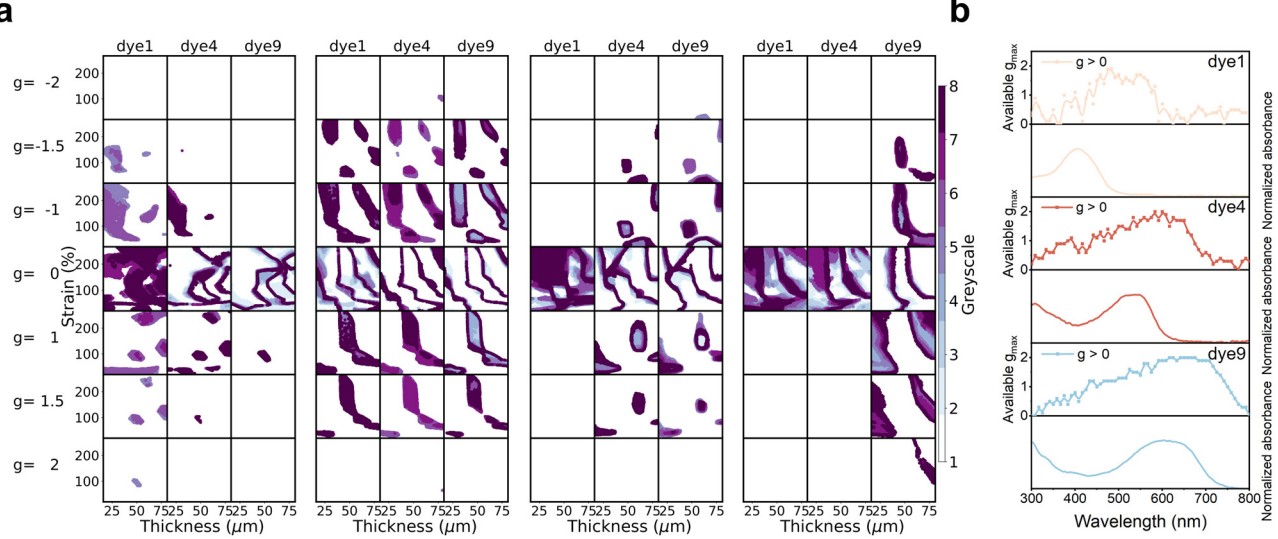

**Fig. 6 | Ability of dyes to modulate g at different wavelengths. a** Distributions of available answers for a target $g_{abs}$ value (from top to bottom, each row represents −2.0, −1.5, −1.0, 0, 1.0, 1.5, and 2.0, respectively) at a given wavelength (from left to right, each table represents 400 nm, 500 nm, 600 nm and 700 nm, respectively) using a particular dye molecule (from left to right, each column represents dye 1, dye 4, and dye 9, respectively in each table). In each block, the x axis, the y axis, and the greyscale represent the thickness, the strain and the concentration of dye molecules, respectively. For simplicity, the fourth structural parameter, stacking angle, was set to +45°. The full details are presented in Supplementary Fig. 16-20. **b** Wavelength dependence of maximum $g_{abs}$ values possible for each dye molecule, with their absorption spectra plotted for comparison.

are therefore more easily accessible than those with high magnitude (close to ±2). When the target is set to ±2, the theoretical limit, recommendations are not always available (such as the cases shown in the left and right blocks at 400 nm (Supplementary Fig. 16) and 700 nm (Supplementary Fig. 20). Another interesting observation is

the wavelength dependence. For a specific dye molecule, the number density of available answers is correlated with its absorption spectrum. For example, the yellow dye molecule can generate a high $g_{abs}$ value near 400-500 nm (Fig. 6b and Supplementary Fig. 21–22). This is reasonable since the absorption of dye molecules in a chiral film can

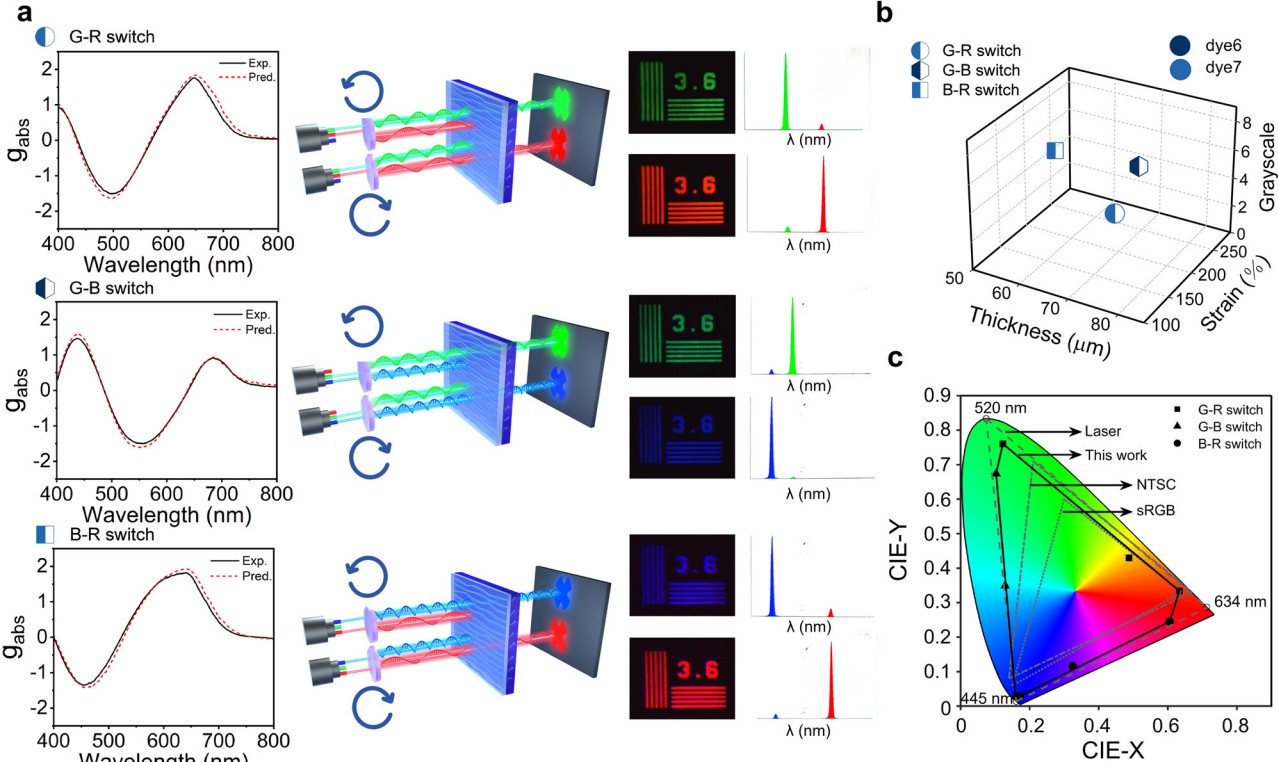

**Fig. 7 | Inverse design of films for circular polarization-switchable multiplex laser display. a** Representative set of parameters that generate the greatest differences in $g_{abs}$ values between two wavelengths. The following color switches are shown from top to bottom: G-R (green-red) switch between 520 and 634 nm; G-B (green-blue) switch between 445 and 520 nm and B-R (blue-red) switch between 445 and 634 nm. The first column is the predicted vs. experimentally measured CD spectra of the three circular polarized (CP)-selective filters. The commercial laser sources available for color switching of laser displays are 445 nm (blue), 520 nm (green), and 634 nm (red), respectively. To perfectly match the wavelengths of the above commercial lasers, the inverse design of the films used as color filters exhibited maximum/minimum values at about 650 nm/495 nm (G-R switch), 550 nm/440 nm (G-B switch), and 460 nm/640 nm (B-R switch) could be obtained

respectively. The filters selectively filter out one of two laser beams according to their circular polarization state, as shown in the cartoon illustration of the optical setup in the second column. The third and fourth columns represent the obtained images and color analysis if the incident light is pure right-handed circular polarization (RCP) or left-handed circular polarization (LCP), respectively. The greater the difference in the $g_{abs}$ values is, the purer the color. **b** 3-D plot of the parameter space of three membranes for the three color switches. **c** CIE1931 chromaticity diagram. The areas enclosed by the solid lines are colors accessible using our three color switches. It covers more than 90% of the maximum possible color space using three monochromatic lasers at 445 nm, 520 nm, and 634 nm. A typical color gamut is shown for comparison.

modulate the relative strength of right circular polarization (RCP) and left circular polarization (LCP). This ability weakens when the target region moves further away from its absorption. In most cases, a given $g_{abs}$ value can be successfully achieved by inverse design at any user-specified wavelength. Figure 5c contains a plot for a series of inversely designed and as-prepared films that can generate $g_{abs} = 1.5@500$ nm (more examples are shown in Supplementary Fig. 23 and Supplementary Table 2). $g_{abs}$ values as high as 1.9 can be usually reached, which is the highest $g_{abs}$ ever reported (Fig. 5d and Supplementary Table 3). In some cases, the output of the reverse model cannot meet our requests. When a very large target $g_{abs}$ value (+2 or −2) is needed at the two ends of the visible spectrum (<400 nm and >700 nm), the model usually fails to provide an effective design and the maximum $g_{abs}$ value obtainable is less than 1.5 (or −1.5 at minimum) by gradually relaxing the target value (full details in Supplementary Fig. 24). This is probably because the target property exceeds the capability of the material. For example, at wavelength of 300 nm, the errors of the reverse network are much larger than those at the center of the visible spectrum (Supplementary Fig. 25). This phenomenon can be attributed to the lower absorption, which results in an error for the $g_{abs}$ value and inherent system error in those regions. In principle, this problem could be alleviated by including more dye molecules with absorption in the UV and near-infrared regions.

## Inverse design of color switches for multiplex laser display

The success of inverse design for membranes with single $g_{abs}$ values at single wavelengths prompted us to further extend our method to multiple $g_{abs}$ values at multiple wavelengths, in response to the demand for real applications. In the first application of circular polarization-based multiplex color laser display (Fig. 7), opposite dissymmetry factors are needed for CP-selective transmission at two of three prime colors. The CP state of light can provide one more degree of freedom to code functionality in laser displays. For example, after passing this CP-selective filter, a laser beam containing two prime colors (green@520 nm and red@634 nm, respectively, as shown in the top case in Fig. 7a), would cast different colors depending on its CP state. In this particular case, a red RCP light and a green LCP light would transmit through (or vice versa, allowing a red LCP light and a green RCP to transmit through, as shown in Supplementary Fig. 26b). If a light with no CP is used, no discrimination occurs against either red or green, and an orange color is produced in overlapping regions (as shown in the central columns in Supplementary Fig. 26a or b). Similarly, the color switch can be accomplished between green@520 nm and blue@445 nm (the center case in Fig. 7a), as well as between red@634 nm and blue@445 nm (the bottom case in Fig. 7a). In each of the three cases, the best filter to differentiate two lasers would be selectively RCP-passing (most positive $g_{abs}$) at one wavelength and

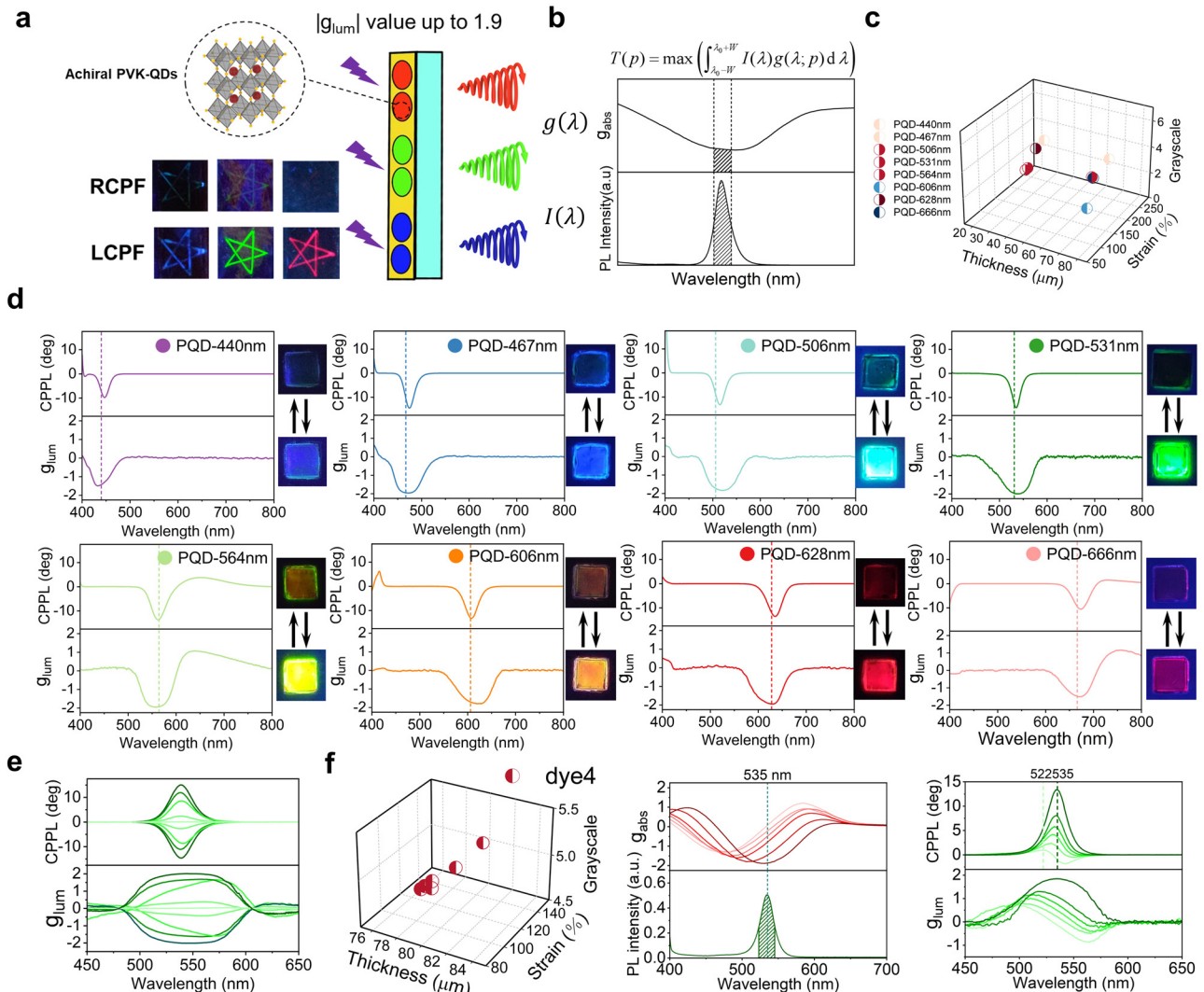

**Fig. 8 | Inverse design and fabrication of chiral film as a universal platform to convert nonpolarized fluorescence into RCP luminescence. a** Scheme of the fluorescent films created from perovskite-based quantum dots (PQDs) and chiral films. The emission spectra of each PQDs are given in Supplementary Fig. 31. **b** Target function ($T$(p), where **p** stands for spectrum embedded descriptors) aims to maximize dissymmetry factor $g_{abs}$ values ($g(\lambda)$), which are weighted by the intensity distribution ($I(\lambda)$) of the fluorescence emission generated by the PQD. **c** Eight sets of parameters generated and recommended by the reverse model, each corresponding to a particular type of PQD. **d** Experimental verification of each of eight recommendations and the accompanying photographs of the composite films under right-handed circularly polarized filter (RCPF) (upper) and left-handed circularly polarized filter (LCPF) (lower). **e** An example of target luminescence dissymmetry factor $g_{lum}$ values tuneable from −2 to +2. **f** Fine-tuning CP luminescence. Left panel: 3D plots of the design parameters of a series of films with off-resonance weight distribution functions. Central panel: The circular dichroism (CD) spectra of the as-prepared films, with the emission spectrum of the PQDs plotted at the bottom for comparison. Right panel: circular polarized photoluminescence (CPPL) with different spectral profiles obtained using these films and the PQD.

LCP-passing (most negative $g_{abs}$) at the other wavelength. The reverse model is employed to generate three recommendations, one for each switch. The three sets of parameters are shown in the 3D plot in Fig. 7b. The difference between the two $g_{abs}$ values for the two prime colors reaches 2.9, 2.6, and 3.1 respectively, while the theoretical limit is 4. The filters can effectively switch laser colors, as reported in the last column of Fig. 7a. The unwanted transmission of light through the chiral film is typically less than 5% of the selected light. With these three one-to-one switches, a full-color laser display can be achieved by switching between CP states of lasers in three prime colors (examples are shown in Supplementary Movie 2). Any color within the area shown in the CIE1931 chromaticity diagram (enclosed by the solid lines in Fig. 7c) can be produced by controlling their CP states. The inverse design approach adopted here helps to increase this accessible color space to approximately 90% of the maximum possible gamut, which is defined as the triangular area (enclosed by the dashed lines), in which

its three vertices are the three monochromatic lights at 445 nm, 520 nm, and 634 nm. This color space covers ~140% of a typical NTSC color gamut found in commercial color displays. For comparison, the color gamut of our previous system without the help of inverse design only reached 57% of the NTSC[26]. The color gamut is almost the largest among all chromaticity maps based on the polarization switched color display (Supplementary Table 4).

## Inverse design of filters for circularly polarized luminescence

In the second application, our goal is to design and fabricate a series of chiral films as a universal platform to convert nonpolarized fluorescence into CP luminescence (Fig. 8a). A series of fluorescent films created from perovskite-based quantum dots (PQDs) were prepared. The X-ray powder diffraction (XRD) patterns of pure PVA and MAPbBr$_3$/PVA are shown in Supplementary Fig. 27a. To further investigate the size and morphology of the MAPbBr$_3$ crystals in PVA,

transmission electron microscopy (TEM) was performed (Supplementary Fig. 27b). The MAPbBr$_3$ QDs are uniformly dispersed in PVA with an average diameter of approximately 30 nm. When excited by UV (365 nm), these PQDs emit strong fluorescence with various colors. The emission of each PQD features a distinct wavelength and spectral width (Supplementary Fig. 28) and its achiral nature was characterized (Supplementary Fig. 29). For each one, the reverse model was exploited to design a corresponding chiral film that matches its color and width. Since the goal was to generate the highest luminescence dissymmetry factor $g_{lum}$ at and near the wavelength of fluorescence, we defined a target function as follows:

$$T(p) = \max\left(\int_{\lambda_0 - W}^{\lambda_0 + W} I(\lambda)g(\lambda;p)d\lambda\right) \tag{1}$$

where $p$ denotes the designed parameters in the spectrum embedded descriptors, $\lambda$ is the wavelength, $g(\lambda;p)$ is the CD spectrum under parameter $p$, $I(\lambda)$ is the normalized emission spectrum of the target PQD, and $\lambda_0$ and $W$ are the peak wavelength and half-peak width of $I(\lambda)$, respectively (Fig. 8b). Therefore, the optimal $g_{lum}$ value is obtained by maximizing the $g_{abs}$ values weighted by the intensity distribution of the fluorescence emission of PQDs. For each PQD, a corresponding target function $T(p)$ is used in the reverse model. The results are summarized in Fig. 8c, in which the 3D plot represents the thickness, strain and greyscale; the color codes the corresponding dye molecule (not the color of fluorescence), and the side of half-filling circles represents whether the twist angle is +45° or −45°. The experimental verification confirms the successful execution of the whole procedure. The corresponding experimental CD spectra of the chiral films are shown in Supplementary Fig. 30. The experimentally measured CPPLs are reported in Fig. 8d, in which the maximum $g_{lum}$ reaches 1.9 in most cases; this value, is almost the highest dissymmetry factor reported for perovskite-based CPL (Supplementary Table 5). Note that the inverse design and experimental realization described here aim to maximize RCP luminescence, and the fluorescence becomes brighter or duller when viewed through left- and right-handed CP filters, respectively. A similar procedure can be carried out to maximize LCP instead, and the results are available in Supplementary Fig. 31. The reverse model can be easily simplified if lower $g_{lum}$ values are needed, as shown in the example of Fig. 8e. Additionally, the CP luminescence generated from the twist-stacking films[45,46] not only exhibited strong $g_{lum}$ but also showed a high figure of merit (FOM) which demonstrated the comprehensive quality of the CPL active materials. We calculated the FOM by multiplying the luminescence dissymmetry factor ($g_{lum}$) and the photoluminescence quantum yield ($\phi$), as shown in Supplementary Table 6. The resulting FOM value was up to 0.41, much higher than that of most reported CPL materials (Supplementary Fig. 32 and Supplementary Table 7).

Furthermore, our approach can be conveniently adopted to fine-tune the center wavelength and spectral profile of CP luminescence originating from the same fluorescent source. This task can be readily achieved by choosing a weight distribution function that is slightly off-resonant with fluorescent light. A series of off-resonance distribution functions were used as target functions and then inversely designed using the reverse model, as shown in the 3D plot in Fig. 8f. The films were prepared accordingly, with their CD spectra shown in the central panel of Fig. 8f. The spectral profile of the fluorescent source is plotted at the bottom for comparison. The center wavelength of CPPL gradually shifts away from resonance and the target function moves more off-resonance, as demonstrated by the final CPPL spectra shown in the right panel of Fig. 8f. Interestingly, the spectral profile of CPPL also changes gradually from Gaussian-like to sigmoid-like.

## Discussion

In summary, this paper reports the first use of robotic all-round AI-Chemist with a computational brain to execute the full cyclic process of discovering and preparing chiral films with target chiroptical performance. The QSSAR is first established between the structural parameters, spectral features and optical activities, and then the design parameters can be inversely "calculated". As a result, flexible films with large chiroptical activities at a designated wavelength or wavelength profile over the full visible spectrum can be prepared on-demand by this AI chemist platform. We believe that the strategy adopted in this work is useful for designing various types of optical materials/devices and even for implementing machine learning computation using optics.

## Methods
### Chemicals
Dye molecules I-XX (erythrosin b, direct blue 2, direct blue 6, direct blue 71, basic orange 2, congo red 9, indigo carmine, sirius yellow, aizendirectgreenbh, reactive blue 4, malachite green, methyl orange, basic fuchsin, direct purple 1, direct red 13, eosin y, amanilskybluer, azophloxine, new coccine, erioglaucinediammonium salt), acrylamide, and methacrylic acid were purchased from Aladdin Reagent, Energy Reagent, and Karma Reagent. The polyvinyl alcohol (PVA) films used in this work were obtained from New Blue Sky Material Industry Co., Ltd. with thicknesses of approximately 17 μm, 30 μm, 48 μm, 60 μm, and 80 μm. The weight-averaged molecular weight was Mw ~43 kg mol$^{-1}$ with a polydispersity of 3.8. Polyethylene (30 μm, 50 μm, 80 μm) was purchased from Yunhang Trading Co., Ltd., and polyvinyl chloride (38 μm, 60 μm) was purchased from Wuluba Trading Co., Ltd. and D4 gel (17 μm, 30 μm, 60 μm, 80 μm) were purchased from Heowns Biotechnology Co., Ltd. All the chemicals and films were used as received without further purification.

### Preparation of polyacrylamide and polymethacrylic acid gel
Acrylamide/methacrylic acid (1 g) and photoinitiator irgacure 2959 (2 mg, 0.5 mol%) were dissolved in 2.5 g water. The resulting solution was poured into a glass mold and polymerized under UV irradiation for 5 h to obtain the hydrogel.

### Synthesis of PQDs/PVA
PQD/PVA films were prepared by a modified method analogous to previous work[47]. For the green PQDs (531 nm), CH$_3$NH$_3$Br (0.6 mmol), PbB$_2$ (0.3 mmol), and PVA (2.1 g) were added to 30 mL deionized water. For the blue PQD (467 nm), CH$_3$NH$_3$Cl (0.4 mmol), CH$_3$NH$_3$Br (0.35 mmol), PbCl$_2$ (0.16 mmol), PbBr$_2$ (0.14 mmol), and PVA (2.1 g) were added to 30 mL deionized water. For the red sample, CsBr (0.25 mmol), CH$_3$NH$_3$I (0.3 mmol), PbBr$_2$ (0.08 mmol), PbI$_2$ (0.2 mmol), and PVA (2.1 g) were added to 30 mL deionized water. The reactants were stirred vigorously at 95 °C for 2 h to dissolve the halides and polymer. Other samples with different colors were fabricated with a mixture of CH$_3$NH$_3$X, PbX$_2$, and CsX (X=Cl, Br, I). To prepare PQDs/PVA composites, spin coating was performed on quartz substrate. The composite layers were dried at 90 °C for 5 min, and the PQD/PVA films were obtained.

### Film preparation and characterization by AI-chemist
The chiral film was constructed with the transparent film (top layer) and the dyed film (bottom layer) in a twisted fashion. The preparation and characterization of chiral thin films by AI-chemist is divided into the following steps. (i) Choice of the matrix materials: PVA, PE, PVC, polyimide gel, polyacrylic gel and D4 gel were selected as candidate matrix materials for transparent films. Among them, films of PVA, PE, and PVC with different thicknesses were commercially available and purchased without further treatment. The D4 gel were prepared by a drop casting method. PVA was used as the dyed film due to its affinity

for dyes. The uniformity of the film thickness was confirmed by scanning electron microscopy (SEM), as shown in Supplementary Fig. 33. (ii) Dyeing a film: AI-Chemist soaked a thin film in the dyeing solution (0.1–10 mg mL$^{-1}$) for a designated dyeing time (0 s to 420 s) at 60 °C to disperse dye molecules in it and generated dyed films with different greyscales depending on the absorption maximum of the dyed films. Due to the different affinities between dyes and PVA, the loading capacity of the dyes varied from 0.1 μmol cm$^{-2}$ to 2 μmol cm$^{-2}$. Additionally, the chiroptical activity of the films was greatly dependent on the absorbance of the dyed films. (iii) Stretching the films: The films were stretched by the AI chemist to a certain strain (20–600%) in a solution containing 2 wt% of boric acid at 60 °C to obtain macroscopically anisotropic films. (iv) Characterization of the linear dichroism (LD) properties: the LD properties were obtained by comparing the transmission of horizontally polarized transmitted light ($T_\parallel$) and vertically polarized transmitted light ($T_\perp$) through the stretched film. LD is defined as the following formula: $\mathrm{LD} = \log_{10}\left(\frac{T_\perp}{T_\parallel}\right)$. The transmittance was measured using a UV–vis spectrometer (Shimadzu UV-2700) with a linear polarizer. (v) Stacking two anisotropic layers in a twisted fashion to construct a heterostructure film: The chiral films were constructed by stacking the transparent film (top layer) and dyed film (bottom layer) in a twisted fashion, in which the top layer acts mostly as a linear birefringence (LB) film and the bottom layer acts mostly as a linear dichroism film. The bottom layer was rotated in either a clockwise (+45°) or counterclockwise (−45°) fashion with respect to the top layer to generate a heterobilayer film. (vi) Characterization of the chiroptical properties of the resulting film: the CD properties were obtained using a UV–vis spectrometer with a circular polarizer[48]. The incident light was converted into circularly polarized light by passing through a linear polarizer, a broadband quarter-wave plate. $g_{abs}$ were obtained by the relative difference in absorption through the sample of left vs. right CP light according to the following formula: $g_{abs}$ ($g_{abs} = (A_L - A_R)/((A_L + A_R)/2)$). The $g_{abs}$ spectra measured by a UV–vis spectrometer were verified by CD characterization. To exclude the possible angle dependent effect, the sample was rotated perpendicular to the optical path of the spectrometer at each step of 45°. Eight measurements at different rotation angles were averaged to obtain the CD response for subsequent analysis (Supplementary Fig. 34), confirming that the obtained chiroptical signals resulted from genuine chiroptical effects intrinsic to the twisted alignment. In addition, the chiral films also exhibited excellent stability after treatment with boric acid. As shown in Supplementary Fig. 35, no obvious degradation could be detected for the prepared chiral stacked films even after storage for 20 days in air. CD characterization was performed by a commercial CD spectrometer (JASCO-1500). Circular polarized photoluminescence was directly measured using a JASCO CPL-300 spectrometer.

## Giant chiroptical activity and modulation mechanism of the heterostructured bilayer

As mentioned in previous work[49], the CD signals of these chiral stacked films can be clarified by Jones matrix presentation. Assuming no nearfield interaction occurs between the two stacking layers, the total transmission Jones matrix of a chiral film is the product of the two matrices representing the two single layers. In Jones matrix presentation, the transmission of an anisotropic layer can be written as follows:

$$T = \begin{bmatrix} |t_1| \cdot e^{i\gamma_1} & 0 \\ 0 & |t_s| \cdot e^{i\gamma_s} \end{bmatrix} = |t_s| e^{\frac{i(\gamma_1 + \gamma_s)}{2}} \begin{bmatrix} |t| \cdot e^{\frac{i\Delta\gamma}{2}} & 0 \\ 0 & e^{-\frac{i\Delta\gamma}{2}} \end{bmatrix}, \quad (2)$$

where $t = t_l/t_s$, $\Delta\gamma = \gamma_l - \gamma_s$. t represents the amplitude tuning related to the anisotropy, and $\Delta\gamma$ represents the phase shifts. In our case, left/right-handed circularly polarized light passes through the transparent

layer (first layer) and then the dyed layer (second layer). The extinction ratio of the transparent layer is approximately 1 and the properties of the transmitted light through the film are largely dictated by the phase difference. At a certain degree of stretching, the transparent layer can function as a quarter wave plate at a certain wavelength. Upon passing through the transparent layer, left-handed and right-handed circularly polarized light are transformed into two orthogonally polarized beams at angles of 45° and −45°, respectively, relative to the optical axis of the transparent film (see the top-left edge in Supplementary Fig. 1a and i). Afterward, the orthogonally polarized light passes through the second layer (dyed film) in perpendicular and parallel orientations to its optical axis (the optical axis of the dyed film is at a 45° angle to the optical axis of the transparent film). The maximum of $g_{abs}$ can be achieved by applying the formula: $g_{abs} = (A_L - A_R)/((A_L + A_R)/2)$ under these circumstances. If the first layer (transparent film) is loaded with dye, polarized absorption occurs in addition to the phase difference, and the polarization effect of the first layer cannot be ignored. In this case, the polarization state of the transmitted light through the first layer would be changed. Therefore, the inter angle between two transmitted light beams would decrease, causing a corresponding decrease in $g_{abs}$ (see the bottom-left edge in Supplementary Fig. 1a and ii).

To better illustrate this issue, we performed simulations using the Jones matrix[49]. We assume $\Delta\gamma_1 = \Delta\gamma_2 = \pi/2$ (the phase difference of the first layer and the second layer), and $t_2 = 10000$ (the extinction ratio of the second layer), and set $t_1$ (the extinction ratio of the first layer) as the variable. $t_1 = 10000$ simulates a homostructured bilayer, whereas $t_1 = 1$ represents a heterostructured bilayer. As shown in Supplementary Fig. 1b, $g_{abs}$ drops quickly with increasing $t_1$. To summarize, we believe that the heterostructured bilayer tends to generate a stronger CD signal in our settings.

## Cluster-screening of structure/process parameters

The two monolayers in the composite films were prescreened individually. For each monolayer film, the maximum LD and the average $T_0$ near the wavelength of the maximum LD were assumed to indicate the performance. Films with high LD and $T_0$ (top 40%) were labeled as high performance, while those with low LD and $T_0$ (bottom 40%) were labeled as low performance. Then, hierarchical clustering based on normalized LD and $T_0$ was performed on films to form several categories. Categories that contain high performance films were regarded as high performance categories, as are for low performance categories. After clustering, the posterior probabilities of each parameter value to construct films in high/low performance clusters were used to judge whether to reject or retain the value. For example, $P_{high}(strain=100\%)$ = (number of films with 100% strain in high performance clusters)/ (number of films with 100% strain in total). After prescreening, 10 of 20 dye molecules were retained and renamed dyes 1–10 (sirius yellow, methyl orange, congo red, direct red 13, direct purple 1, amanilsky-bluer, direct blue 2, direct blue 6, direct blue 71, and aizendirect-greenbh). When the dataset for forward prediction was constructed, several points in the dyeing time (0–200 s) corresponding to the absorption intensity of the dyed films were sampled and transformed into greyscale (marked as 1–8).

## ML models

**Neural networks.** The forward network is a fully connected neural network consisting of a 125-node input layer, a 121-node output layer, and 5 hidden layers with 200, 250, 250, 250, and 200 nodes. Batch normalization and the ReLU activation function are applied to the input layer and each hidden layer. Fivefold cross-validation is applied to the whole dataset to optimize the division of the training set and testing set. Before training, the size of the training set is extended 10-fold by adding noise with a scale of 0.003. The SmoothL1Loss is used as the loss function and an Adam optimizer is used to update the

network weights[50]. The forwards network is trained for 10000 epochs with hyperparameters including batch size (256), and learning rate (0.001).

In the stage of inverse design, the parameters of the forward network remain unchanged so that only the generator is trained. The loss is calculated based on the differences between the predicted and target properties, and then the gradients of the parameters of the generator are calculated through back propagation. The generator is a fully connected neural network consisting of a 20-node input layer, a 5-node output layer, and 3 hidden layers of 40, 80, and 20. All settings of the generator are the same as the forward network except for the batch size (16) and the number of epochs (50). To connect the generator with the forwards network, the first number of the generator's output, which represents the species of dyes, is converted into a one-hot vector, and then multiplied with a $10 \times 121$ matrix where each row represents the absorption spectrum of one dye. As a result, the species of dyes are transformed into the corresponding absorption. Both the forward and inverse models were implemented using PyTorch[51].

**Dataset**. A total of 1493 samples were selected from the $10^5$ possible combinations of the selective absorption of dye molecules and four structure/process parameters. Among them, the dyes (10 types), thickness (17 μm, 30 μm, 48 μm, 60 μm, 80 μm), and stacking angle (45°, −45°) were finalized after the second round of screening. The strain (20%, 40%, 53%, 67%, 80%, 100%, 120%, 133%, 167%, 200%, 233%) and the greyscale, which was determined by dyeing time (0–200 s), were subsequently fixed at specific intervals. The training dataset was selected semi-randomly from all these possible combinations of experimental parameters. Approximately 23 binary combinations of thickness and strain were chosen randomly. For each thickness-strain combination, 4–8 greyscales were tested. In addition, 7–9 tertiary combinations of thickness-strain-greyscales were randomly selected which are different from the above 24 binary combinations. This results in a total of approximately 150 thickness-strain-greyscale combinations. The same tertiary combinations were repeated for each of the 10 dyes, as shown in Supplementary Fig. 36. This was performed to guarantee a more even distribution of the training set in the parameter space, thereby increasing its representativeness. In addition, this helped the robotic chemist plan. In the end, data from 1493 samples were successfully collected, and the collection formed the training set used in this work.

**Hyperparameters**. The hyperparameters were tuned via the random search method[52], the performance of the model was greatly affected by changes in the learning rate, and good results were obtained between 0.0001 and 0.08. However, when the learning rate was set to 0.09 or larger, the model significantly deteriorated. The results were not very sensitive to epoch size or batch size if they were not very small. For example, the results of the reverse model trained under 300 epochs and a 0.0001 learning rate are virtually the same as those of the model trained under 50 epochs and a 0.001 learning rate.

### Reporting summary
Further information on research design is available in the Nature Portfolio Reporting Summary linked to this article.

## Data availability
The dataset supporting the findings of this study has been deposited in the Github repository[53], and is available at https://doi.org/10.5281/zenodo.8198186. Source data are provided in this paper.

## Code availability
The codes that support the findings of this study have been deposited in the Github repository[53], and are available at https://doi.org/10.5281/zenodo.8198186.

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

## Acknowledgements

This research was carried out with funding from the Strategic Priority Research Program of the Chinese Academy of Science (XDB0450302), National Natural Science Foundation of China (22071233, 52373122, 22025304, 22033007), the Basic Research Fund for the Central Universities (WK3450000006), the CAS Project for Young Scientists in Basic Research (YSBR-005) and AnHui Estone Materials Technology Co., Ltd. This work was partially carried out at the University of Science and Technology of China's Center for Micro and Nanoscale Research and Fabrication. We thank the Hefei advanced computing center and USTC supercomputing center for providing computational resources for this project.

## Author contributions

Y.X., S.F., G.Z., X.C., J.J. initiated the work and drafted the manuscript. Y.X., S.F., G.Y., W.Z., Q.Z. developed the workflow, and implemented the robot positioning approach, wrote the control software, designed the measurement station. Y.X., L.G., and Z.L. prepared the samples and performed the spectroscopic experiments. Y.X., L.D., and P.Y. performed the optical experiments. S.F., A.C., Z.J., and W.D. performed the model construction and calculations. L.W., J.L., C.Z., Z.F., Z.Z., and L.X. acquired and evaluated the spectroscopic ellipsometry data. All authors discussed the results and contributed to the writing and editing of the manuscript.

## Competing interests

The authors declare no competing interests.
