## [Peer Review file · Nature Communications]

REVIEWER COMMENTS

Reviewer #1 (Remarks to the Author):

Review Assignment

Manuscript Title: Inverse design of chiral functional film by a robotic AI-Chemist

Venue: Nature Communications

PI: Gang Zou - University of Science and Technology of China, Hefei

Decision: Revisions necessary

Summary: "Inverse design of chiral functional film by a robotic AI-Chemist" by Xie et al reports on the use of AI-guided robotic chemist to prepare chiral films with desired chiroptical activities. The authors were able to use data from the automated experiments to design a forward machine learning model to establish quantitative structure-spectrum-activity relationship for chiroptical films. Additionally, they were able to design an inverse AI model utilizing spectrum embedded descriptors. The authors were able to use this model to find structural and process parameters for films with any desired chiroptical properties. They found a chiroptical film with the highest ever gabs value of 1.9 reported in literature. Finally, the inverse design model and robot were used to design and fabricate films for two applications including color filters for laser display and conversion of non-polarized fluorescence into circularly-polarized luminescence.

Claim 1: The authors developed a forward prediction algorithm capable of predicting the spectrum based on the input conditions

The authors developed a forward prediction model to predict the CD spectra of chiroptical films. The neural network performed very well with mean absolute error, root mean squared error (RMSE) and coefficient of determination (R2) of prediction model on the testing set are 0.04, 0.06, and 0.985, respectively (Supplementary Figure 6). The model is also tested against 20 new sets of experiments and the predicted structural/process properties were in excellent agreement with the experiments (Supplementary figure 7-10).

Claim 2: The authors developed an inverse-design algorithm capable of suggesting input conditions that result in films with a target g-value

The authors demonstrate the accuracy and utility of their inverse design algorithm in figure 2c by showing that the model suggests four films with a target of $g=1.5$ at 500 nm and predicts their spectrums with a high degree of accuracy.

Claim 3: The authors are capable of producing films with user-specified properties using the prediction algorithms and AI-chemist.

The authors demonstrate this by designing and fabricating films for (1) filtering color in laser display (as demonstrated in figure 3) and (2) converting non-polarized fluorescence into circularly-polarized luminescence (as demonstrated in figure 4).

Decision: This manuscript presents an impressive body of work describing an AI-driven materials science platform dedicated to automated design and fabrication of chiroptical thin films applicable for optics, catalysis, and chiral sensing. The inverse design of chiroptical films with defined properties would likely be of interest to the readership of Nature Communications. However, the poor writing and organization within the document detracts from the overall quality of the work. The manuscript contains many grammatical errors and generally lacks scientific writing style. It is poorly organized, with important information that belongs in the main text and methods found only in the supplementary materials or figure captions. I would recommend the publication of this manuscript in Nature Communications, only after the writing and formatting are significantly improved.

Comments:

Grammatical error in title. Proper grammar would be films with an “s” since the authors are likely referring to the many films they fabricate throughout the study.

Line 16, It is not clear in the abstract if the gabs value of 1.9 is the highest one ever achieved in the literature. Authors need to give some comparison

Please describe g value. It is first mentioned in the abstract but also mentioned on line 172 and never defined.

Line 19, CP is not defined

Line 32, what is “assembles”? I believe it may be a typo.

Line 44-46, the sentence is abruptly ended

Line 62 CD is not defined

Line 124, homo should be “hetero”

Extended Figure data 1-e, the concentration of the dyes are not mentioned

Consider making it explicit what the inputs and outputs are for the materials synthesis and models. It is not entirely clear. A list is given on line 45, line 76, and supplementary table 1, however those variables seem to be different from the ones used as inputs to the model as described in Fig 1 and supp fig 6.

Line 101- 104, It is not clear to me what are the exact experimental absorption spectra and structure/process parameters involved in this study which are fed into the neural network algorithms. What is the exact dimension of their dataset? Is it $10^5 \times 1493$?

Can the author provide supplementary data on how the machine learning model is trained?

Please include AI techniques in the methods section. Consider moving the methods mentioned in supplementary figures (such as the details in the supp fig 6 caption) to the methods section along with more details.

Line 84 “ An AI-chemist is recruited to solve this problem”. It is not clear if the robot has been built for the current work or if few other scientific work have been carried out before by utilizing the AI-chemist. If yes, authors should cite them. “An all-round AI-Chemist with a scientific mind” by Zhu et al. (National Science Review, 2020) is the one closest example which I can find and has some of the same authors and is from the same university.

Could the authors provide more detail on how the films were made?

Line 99 to 101. Cluster analysis should be defined somewhere in SI or method section and needs to be referenced where it was first mentioned in the main text

Line 109: Abrupt starting of “Customized Manufacture”

Line 106-108, it is not clearly mentioned what was the motivation to try inverse-design methods and the method is not cited as well. Did the author develop those methods?

Line 142. , replace “which” to “where”

Line 166-167, How did the AI-chemist choose 1493 combinations from 10^5 ? Could the authors provide more details on the training data?

How did the author tune the hyperparameters of the forward and reverse neural network model?

Line 199, “If successful,....” This line is not clear.

Line 241, Figure caption “ Some of them fall short to the target” What do authors try to convey from this statement?

The R2 value for the reverse model is 0.95. Is there any particular set of data points where the inverse model failed to predict sets of parameters? If yes, authors should provide an example and why the model failed.

Line 220, RCP and LCP are not defined in the manuscript

Line 222, grammatical errors in the sentence

Line 287, grammatical error

Line 318-320, grammatical error

Supplementary figure 7 missing “angle” as a parameter which is an input to the neural network

The authors could improve the text by calling out methods for experimental procedures. For example, when they say they performed clustering, they could say “see methods - clustering” and then describe it. Same goes for physical processes such as film synthesis.

Supplementary fig 20 references 2e in the main text but that does not exist

Figure 2C: It would be beneficial for the author's to provide a table in the SI of the suggested recipes based on the input so the reader can get an idea of what the model is outputting

Reviewer #2 (Remarks to the Author):

This manuscript by Xie et al. used a robotic “AI-chemist” to fabricate and characterize twisted stacked chiral thin films with a multitude of design/process parameters in an automated and high-throughput fashion, and high-performance chiral thin films could be designed/fabricated on-demand and used for polarization-selective laser display and visible CPL emission upon quantum dot doping. The authors did an excellent job of combining robotic experimentation, ML-based inverse design, and chiroptical applications using high-performing designs in the visible range. The combination thereof is sufficiently novel and is critical for advancing the research on chiral materials for optical/photonic applications. In particular, the g-factor of the twisted stacked materials could be as high as 1.9, which being the highest value ever reported represents a significant advance in itself.

Based on the above considerations, I enthusiastically recommend its publication in Nature Communications after the authors address the following minor points and provided that an expert of AI and/or robotic experimentation (that I am not) also evaluates the manuscript.

1. While the manuscript is overall well-written, there are some typos to be corrected, as well as stylistic or word choices the authors should reconsider and rewrite. These include but are not limited to the following:
 - a. “optima” in the caption of Figure 1 and Supplementary Video 1;
 - b. The last sentence of the first paragraph (Lines 109-111);
 - c. “bad points” reads funny and uninformative (Line 146)
2. The dyes are used for selective absorption. As such, the basic optical properties of the dyes should be characterized.
3. For dye loading, what is the loading capacity of the dyes? Could the authors provide a quantitative value/estimate? One could argue that this will be different for different matrix-dye combinations, thus affecting the chiroptics of the resultant chiral films. Could the authors comment on this?
4. Could the authors provide the LD spectra for 2-3 stretched thin films under different strains? These should be done with rotated samples.
5. The authors should provide cross-sectional SEM images confirming the thickness of the stacked thin films and ensure its relative spatial homogeneity.

6. The CD spectra in this work are all generated using UV-vis measurements equipped with an external circular polarizer. I presume the CD spectrum is then a calculated one, whereas most works use a commercial CD spectrophotometer that directly reports ellipticity values. I would like the authors re-measure CD spectrum of several chiral thin films on a commercial CD spectrometer to validate their results.
7. Could the authors also provide the CD spectra of a chiral thin film by rotating the chiral films under normal incidence and also the averaged CD spectra? There are typically slight variations when doing this, and the averaged value should be reported.
8. Per Extended Fig. 1, could the authors explain why heterobilayer stacking gives stronger CD response than homobilayer stacking? At least your previous work suggests similar or even decreased CD for twisted heterobilayers (Adv. Optical Mater. 2022, 10, 2102197).
9. The use of twisted stacked materials as a platform for CPL emission has been previously reported. The authors should consider reference some prior works in this area (e.g., Adv. Mater. 2022, 2209539) and discuss accordingly. For example, what is the quantum yield ϕ ? The authors should calculate the FOM = $\phi \times g_{lum}$ and compare it to previous works.
10. The basic materials characterization of the synthesized perovskite QDs is missing, e.g., TEM images with size distribution. Its achiral nature should also be ascertained (e.g., CD of a thin film of perovskite QDs). For Supplementary Figure 23, what is the excitation wavelength used? Are these normalized values?
11. Methods: the names of the companies from which dye molecules and polymer films are purchased should be provided. In particular, the nominal physical-chemical information (e.g., the degree of polymerization, polydispersity index, any additives/plasticizers) of the polymer films should be cited and best if characterized in-house by the authors. Importantly, could the authors comment on whether or not the chiroptical performance of the chiral films depends on these polymer characteristics?
12. One biggest concern I have for the current results is the stretching protocol used. It is very clear that the mechanical strain induces 1D anisotropy in the polymer film, and the strain used is a key determinant of the ultrahigh g -factor. But what if the film relaxes over time in the short run? What about polymer fatigue over time in the long run? How would the chiroptics of the twisted stacked films be affected?
13. Could the authors confirm the stability (both physical and chemical) of the twisted stacked chiral thin films over time?

REVIEWER COMMENTS

Reviewer #1 (Remarks to the Author):

Review Assignment

Manuscript Title: Inverse design of chiral functional film by a robotic AI-Chemist

Venue: Nature Communications

PI: Gang Zou - University of Science and Technology of China, Hefei

Decision: Revisions necessary

Summary: "Inverse design of chiral functional film by a robotic AI-Chemist" by Xie et al reports on the use of AI-guided robotic chemist to prepare chiral films with desired chiroptical activities. The authors were able to use data from the automated experiments to design a forward machine learning model to establish quantitative structure-spectrum-activity relationship for chiroptical films. Additionally, they were able to design an inverse AI model utilizing spectrum embedded descriptors. The authors were able to use this model to find structural and process parameters for films with any desired chiroptical properties. They found a chiroptical film with the highest ever gabs value of 1.9 reported in literature. Finally, the inverse design model and robot were used to design and fabricate films for two applications including color filters for laser display and conversion of non-polarized fluorescence into circularly-polarized luminescence.

Claim 1: The authors developed a forward prediction algorithm capable of predicting the spectrum based on the input conditions

The authors developed a forward prediction model to predict the CD spectra of chiroptical films. The neural network performed very well with mean absolute error, root mean squared error (RMSE) and coefficient of determination (R²) of prediction model on the testing set are 0.04, 0.06, and 0.985, respectively (Supplementary Figure 6). The model is also tested against 20 new sets of experiments and the predicted structural/process properties were in excellent agreement with the experiments (Supplementary figure 7-10).

Claim 2: The authors developed an inverse-design algorithm capable of suggesting input conditions that result in films with a target g-value

The authors demonstrate the accuracy and utility of their inverse design algorithm in figure 2c by showing that the model suggests four films with a target of $g=1.5$ at 500 nm and predicts their spectrums with a high degree of accuracy.

Claim 3: The authors are capable of producing films with user-specified properties using the prediction algorithms and AI-chemist.

The authors demonstrate this by designing and fabricating films for (1) filtering color in laser display (as demonstrated in figure 3) and (2) converting non-polarized fluorescence into circularly-polarized luminescence (as demonstrated in figure 4).

Decision: This manuscript presents an impressive body of work describing an AI-driven materials science platform dedicated to automated design and fabrication of chiroptical thin

films applicable for optics, catalysis, and chiral sensing. The inverse design of chiroptical films with defined properties would likely be of interest to the readership of Nature Communications. However, the poor writing and organization within the document detracts from the overall quality of the work. The manuscript contains many grammatical errors and generally lacks scientific writing style. It is poorly organized, with important information that belongs in the main text and methods found only in the supplementary materials or figure captions. I would recommend the publication of this manuscript in Nature Communications, only after the writing and formatting are significantly improved.

Reply:

We thank the referee for the positive opinion and these insightful comments which help improve the quality of this manuscript significantly.

Comments:

1. Grammatical error in title. Proper grammar would be films with an “s” since the authors are likely referring to the many films they fabricate throughout the study.

Reply:

Thanks for referee’s helpful suggestions. We have revised the term ‘film’ with ‘films’ in the title accordingly.

2. Line 16, It is not clear in the abstract if the g_{abs} value of 1.9 is the highest one ever achieved in the literature. Authors need to give some comparison. Please describe g value. It is first mentioned in the abstract but also mentioned on line 172 and never defined.

Reply:

Thanks for referee’s reviews and helpful suggestions. We have defined the g_{abs} “the dissymmetry factor” in the abstract (line 16) and added the text on page 3, lines 74-76 according to the referee’s suggestions, as shown below:

“The chiroptical response was described using the absorption dissymmetry factor g_{abs} ($g_{abs}=(A_L-A_R)/((A_L+A_R)/2)$), where A_L , A_R represent the absorption of left-handed and right-handed CP-light.”

We have revised the text on page 1, lines 15-16 of the abstract according to the referee’s suggestions, as shown below:

“A series of chiral films with dissymmetry factor as high as 1.9 ($g_{abs}\sim 1.9$) had been identified out of more than 100 million possible structures.”

*In addition, we included the comparison of the dissymmetry factor g_{abs} ever reported in the literature with our own work in **Supplementary Table 3**. Our results demonstrate that the value of 1.9, to the best of our knowledge, is almost the highest one ever achieved in the literature using polymeric or organic materials.*

3.

- a) Line 19, CP is not defined
- b) Line 32, what is “assembles”? I believe it may be a typo.
- c) Line 44-46, the sentence is abruptly ended
- d) Line 62 CD is not defined

e) Line 124, homo should be “hetero”

Reply:

We thank the referee for this comment and have revised the text according to the referee's suggestion.

- a. *We have defined CP when it is first mentioned “circularly polarized (CP)” on page 1, line 18 in the abstract.*
- b. *We have revised the “assembles” to “assemblies” in the introduction, on page 2, line 31.*
- c. *We have revised the sentence “In our chiral stacked system, numerous spectral and structural parameters significantly affect the resulting chiroptical properties. These parameters include materials selection, thickness and strain of transparent films, absorption, thickness, strain and grayscale of dyed films, as well as the twist angle between transparent film and dyed film” on page 2, lines 42-46.*
- d. *We have defined “CD” when it is first mentioned “circular dichroism (CD)” on page 3, line 63.*
- e. *We have revised “homo-structured bilayer” with “hetero-structure bilayer” in the caption of **Extended data Fig.1**.*

4. Extended Figure data 1-e, the concentration of the dyes are not mentioned.

Reply:

*Thanks for pointing out the omission. According to Lambert-Beer's law, there exists a positive correlation between the concentration of dye in the film and its absorption characteristics, which we have thoroughly characterized. We therefore opted to use the maximum absorption of the films in order to more accurately depict the concentration of the dyes. The caption of the **Extended data Fig.1e** is now revised to “Intensity of CD spectra correlates strongly with absorption maximum of the films at 618 nm (i:0.10, ii:0.14, iii:0.18, iv:0.26, v:0.35, vi:0.38).”*

5. Consider making it explicit what the inputs and outputs are for the materials synthesis and models. It is not entirely clear. A list is given on line 45, line 76, and supplementary table 1, however those variables seem to be different from the ones used as inputs to the model as described in Fig 1 and supp fig 6.

Reply:

*Thanks for the referee's helpful suggestions. We are sorry for the confusing expression. The hetero-structured bilayer films are composed of a transparent film (the top layer) and a dyed film (the bottom layer). The spectral and structural parameters of our chiral stacked system include materials selection, thickness, strain of the transparent films, absorption, thickness, strain and grayscale of the dyed films, as well as the twist angle between the transparent film and dyed film before the prescreening. And then, after the second round screening, polyvinyl alcohol (PVA) was selected as the material for the transparent films, the thickness and strain of the dyed films were set to be 80 μm and 400%, respectively (as shown in **Supplementary Table 1**). Thus, only five structure/process parameters (including thickness, strain of transparent films, selective absorption and grayscale of the dyed films as well as twist angle between the transparent film and dyed film) were used as inputs to the model, as described in **Figure 1 and Supplementary Fig.9** (previously **Supplementary Fig.6**).*

We have revised the discussion on page 2, lines 42-46 to depict the parameters before cluster-

screening according to the referee's valuable suggestions, as shown below:

"In our chiral stacked system, numerous spectral and structural parameters significantly affect the resulting chiroptical properties. These parameters include materials selection, thickness and strain of transparent films, absorption, thickness, strain and grayscale of dyed films, as well as the twist angle between transparent film and dyed film."

We have revised the discussion on page 3, lines 81-85 to depict the parameters before cluster-screening according to the referee's helpful suggestions to make it clearer, as shown below:

*"There are several variable parameters, including matrix materials selection, thickness and strain (or degree of stretching related to anisotropy) of transparent films, selective absorption of dye molecules (**Supplementary Fig. 2**), thickness, strain, and grayscale of the dyed films, as well as the twist angle between transparent film and dyed film (**Fig. 1a**)."*

We have added the discussion on page 4, lines 108-111 according to the referee's helpful suggestions to make it clearer, as shown below:

*"Therefore, a portion of the parameters were screened out and only five structure/process parameters were finalized. This resulted in a total number of combinations on the order of 10^5 (see details in **methods-cluster-screening, Supplementary Table 1**)."*

We have added the discussion on page 8, lines 169-175 to depict the parameters after the second round of prescreening which were used as inputs to the model according to the referee's helpful suggestions, as shown below:

"Subsequently, the second round of prescreening further eliminated the films with either low LD or low T_0 . PVA was selected as the material for the transparent films, and the thickness and strain of dyed films were set to be 80 μm and 400%, respectively. Thus, only the spectral parameters (the absorption of dye molecules in the dyed films) and four structure/process parameters (including thickness, strain of transparent films, grayscale of dyed films, as well as the twist angle between transparent film and dyed film) were used as inputs to the model."

6. Line 101- 104, It is not clear to me what are the exact experimental absorption spectra and structure/process parameters involved in this study which are fed into the neural network algorithms. What is the exact dimension of their dataset? Is it $10^5 \times 1493$?

Reply:

*Thanks for the referee's valuable suggestions. The exact experimental absorption spectra of the dyes (I-XX) have been characterized (**Fig. R1**). As mentioned above, five structure/process parameters were used after the prescreening, and the total number of combinations was on the order of 10^5 as shown in **Supplementary Table 1**. In our experiments, 1493 combinations are chosen from 10^5 possible ones. Therefore, the dimension of dataset is 125×1493 , where 125 is the total length of the spectrum embedded descriptors. The details are shown in **methods-dataset**.*

We have added the following figure in the supplementary information as **Supplementary Fig. 2** and revised the corresponding expression on the page 4, line 111-117 to make it clearer, as shown below:

*"Secondly, the AI-chemist fabricated 1493 films by selecting from $\sim 10^5$ possible combinations of these parameters (see **methods-dataset**). The CD spectra of these films were subsequently measured. The spectral parameters (absorption of dye molecules) and four structural/process parameters (construction of film) were combined into spectral embedded descriptors, which*

was used as the input for the forward model to establish the quantitative structure-spectrum-activity relationship (QSSAR).”

Fig. R1. The absorption spectra of 20 dyes (I: erythrosine, II: direct blue 2, III: direct blue 6, IV: direct blue 71, V: basic orange 2, VI: congo red, VII: indigo carmine, VIII: Sirius yellow, IX: aizendirectgreenbh, X: reactive blue 4, XI: malachitegreen, XII: methyl orange, XIII: basicfuchisin, XIV: direct purple 1, XV: direct red 13, XVI: eosin y, XVII: amaniskybluer, XVIII: azophloxine, XIX: new coccine, XX: erioglaucinediammoniumsalt).

7. Can the author provide supplementary data on how the machine learning model is trained? Please include AI techniques in the methods section. Consider moving the methods mentioned in supplementary figures (such as the details in the supp fig 6 caption) to the methods section along with more details.

Reply:

Thanks for the referee’s serious reviews and helpful suggestions, the supplementary data about how the machine learning model was trained has been described in the method section.

We have added and supplemented the discussion in the method section, on page 25, lines 562 to 585 according to the referee’s helpful suggestions, as shown below:

“Neural networks: The forward network is a fully connected neural network consisting of a 125-nodes input layer, a 121-nodes output layer, and 5 hidden layers whose numbers of nodes are 200, 250, 250, 250, and 200, respectively. Batch normalization and the ReLU activation function are applied on the input layer and each hidden layer. 5-folds cross-validation is applied on the whole dataset to optimize the division of training set and testing set. Before training, the amount of training set is extended 10-folds by adding noise with the scale of 0.003. The SmoothL1Loss is used as the loss function and an Adam optimizer is used to update the network weights⁵⁰. The forward network is trained for 10000 epochs with the hyper-parameters including batch size (256), and learning rate (0.001).

In the stage of inverse design, the parameters of forward network are kept unchanged and so

that only the generator is trained. The loss is calculated based on the differences between the predicted and target properties, and then the gradients of parameters of generator are calculated through back propagation. The generator is a fully connected neural network consisting of a 20-nodes input layer, a 5-nodes output layer, and 3 hidden layers with 40, 80, and 20 nodes, respectively. All settings of the generator are the same as the forward network except the batch size (16) and the number of epochs (50). To connect the generator with the forward network, the first number of the generator's output, who represents the species of dyes, is converted into a one-hot vector, and then multiplies with a 10×121 matrix where each row represents the absorption spectrum of one dye. As the result, the species of dyes are transformed into the corresponding absorption. Both the forward and inverse models were implemented using PyTorch⁵¹."

*The following reference have been added to the **Main Manuscript**:*

50. Kingma, D. P. & Ba, J. Adam: A method for stochastic optimization. *arXiv*. 1412.6980. (2014).
51. Paszke, A., Gross, S., Massa, F., Lerer, A., Bradbury, J., Chanan, G., ... & Chintala, S. Pytorch: An imperative style, high-performance deep learning library. *Adv. Neural Inf. Process. Syst.* **32**, 8026-8037 (2019).

8. Line 84 "An AI-chemist is recruited to solve this problem". It is not clear if the robot has been built for the current work or if few other scientific works have been carried out before by utilizing the AI-chemist. If yes, authors should cite them. "An all-round AI-Chemist with a scientific mind" by Zhu et al. (National Science Review, 2020) is the one closest example which I can find and has some of the same authors and is from the same university.

Reply:

Thanks for the referee's serious reviews and helpful suggestions, the robot used in our work is adapted from that used in the work of Zhu et al. (National Science Review, 2020).

We have referenced this work in our manuscript as Ref. 27 and added the description on page 4, lines 89 to 92 according to the referee's helpful suggestions, as shown below:

"In our previous work, we developed an all-round AI-Chemist capable of conducting electrocatalysts and photocatalysts experiments²⁷. Herein, the AI-chemist was recruited and adapted to solve this problem."

*The following reference have been added to the **Main Manuscript**:*

27. Zhu, Q. et al. An all-round AI-Chemist with a scientific mind. *Natl. Sci. Rev.* **9**, nwac190 (2022).

9. Could the authors provide more detail on how the films were made?

Reply:

Thanks for the referee's serious reviews and helpful suggestions. The chiral film was constructed by the transparent film (top layer) and the dyed film (bottom layer) in a twist fashion. The preparation of chiral thin films is divided into the following steps. (i) The PVA, PE, PVC, polyimide gel, polyacrylic gel and D4 gel were selected as candidate matrix materials for transparent films. Among them, the films of PVA, PE, PVC with different thickness were

commercially available and purchased without further treatment. The polyimide gel, polyacrylic gel and D4 gel were prepared by the drop casting method. PVA was used as the dyed film due to its affinity for dyes. The uniformity of film thickness was confirmed by scanning electron microscope (SEM). (ii) AI-Chemist soaks a thin film which was in the dyeing solution (0.1-10 mg/mL) for a designed dyeing time (0 s to 420 s) at 60°C to disperse dye molecules in it and achieve the dyed films with different grayscale depend on the absorption maximum of the dyed films. (iii) The films were stretched by AI chemist to a certain strain (20%-600%) in a solution contain 2wt% of boric acid at 60°C to obtain macroscopically anisotropic. (iv) AI-chemist characterizes the LD properties of the stretched films which was described in detail in the methods. (v) The chiral films were constructed by stacking the transparent film (top layer) and dyed film (bottom layer) in a twist fashion, where the top layer acts mostly as a linear birefringence (LB) film and the bottom layer acts mostly as a linear dichroism (LD) film. The bottom layer was rotated in either a clockwise (+45°) or counter-clockwise (-45°) fashion with respect to the top layer to generate a hetero-bilayer film. (vi) The chiroptical properties of the result film was characterized by AI chemist and the detailed process is described in the methods. Above description are supplemented in the method section, on page 20, lines 464 to 507 according to the referee's helpful suggestions.

10. Line 99 to 101. Cluster analysis should be defined somewhere in SI or method section and needs to be referenced where it was first mentioned in the main text

Reply:

Thanks for the referee's comments. The cluster analysis was defined in the method section, we have revised "Prescreening of structure/process parameters" with "Cluster-screening of structure/process parameters" in the method section.

We further supplemented the cluster-screening in the method section, on page 24, lines 547 to 552, as shown below:

"Films with both high LD and T_0 (top 40%) are labeled as high performance, while those with both low LD and T_0 (bottom 40%) are labeled as low performance. Then, hierarchical clustering based on normalized LD and T_0 is performed on films to form several categories. Categories that contain high performance films are regarded as high-performance categories, and so as for low performance ones."

In addition, we added the text "see details in **methods-cluster-screening, Supplementary Fig. 4-5 and Supplementary Table 1**" on page 8, line 176 to 177 (where it was first mentioned) according to the referee's helpful suggestions.

11. Line 109: Abrupt starting of "Customized Manufacture"....

Reply:

Thanks for the referee's comments. To provide a detailed explanation of **Fig. 1c**, we have revised the last sentence of the first paragraph as follows on page 5, lines 121 to 125:

*"Furthermore, the inverse design approach enables the customization of target chiroptical properties, allowing the AI-chemist to identify optimal chiroptical films with target functionalities in two practical applications - circular polarization-selective color filters for multiplex laser display and switchable CP luminescence (as demonstrated in **Fig. 1c**)."*

12. Line 106-108, it is not clearly mentioned what was the motivation to try inverse-design methods and the method is not cited as well. Did the author develop those methods?

Reply:

Thanks for the referee's helpful suggestions. We have cited a reference and added some description in page 4, line 117 to 120, as shown below:

“Lastly, the richness in design space makes traditional forward design approaches based on trial-and-error very inefficient. Machine learning-based QSSAR allows us to develop a reverse model for inverse design, which is adapted from the generative adversarial model⁴⁴, and consist of a generator and a discriminator.”

*The following reference have been added to the **Main Manuscript**:*

44. Ian J. Goodfellow et al. Generative Adversarial Networks, arXiv:1406.2661 (2014).

13. Line 142. replace “which” to “where”

Reply:

Thanks for the referee's helpful suggestions. We have revised the “which” with “where” according to the referee's suggestions on page 7, line 160.

14. Line 166-167, How did the AI-chemist choose 1493 combinations from 10^5 ? Could the authors provide more details on the training data?

Reply:

Thanks for the referee's helpful suggestions. The AI-chemist fabricated 1493 films, which are chosen from $\sim 10^5$ possible combinations.

According to the referee's suggestion, we have added the discussion about the detailed methods to choose 1493 combinations from 10^5 possible combinations of the five structure/process parameters in method section, on page 26, line 586-602, as shown below:

*“**Dataset.** 1493 samples were selected from the 10^5 possible combinations of the selective absorption of dye molecules and four controlled parameters. Among them, the dyes (10 types), the thickness (17 μm , 30 μm , 48 μm , 60 μm , 80 μm) and the stacking angle (45°, -45°) were finalized after the second round of screening. The strain (20%, 40%, 53%, 67%, 80%, 100%, 120%, 133%, 167%, 200%, 233%) and the grayscale, which was determined by dyeing time (0-200 s), were subsequently fixed at specific intervals. The training dataset is selected semi-randomly from all these possible combinations of experimental parameters. About 23 binary combinations of thickness and strain are chosen randomly. For each thickness-strain combination, 4-8 grayscales were tested. In addition, 7-9 tertiary combinations of thickness-strain-grayscales were randomly selected which are not same as the above 23 binary combinations. This results in total about 150 thickness-strain-grayscale combinations. The same tertiary combinations were repeated for each of the 10 dyes, as shown in **Supplementary Fig.36 (Fig. R2)**. This is done to guarantee a more evenly distribution of the training set in the parameter space, thereby increasing its representativeness. In addition, this also facilitated the planning of the robotic chemist. In the end, data of 1493 samples were successfully collected, and the collection formed the training set used in this work.”*

Fig. R2. The dataset of 1493 samples.

15. How did the author tune the hyperparameters of the forward and reverse neural network model?

Reply:

Thanks for the referee’s helpful suggestions. We have added the discussion about the hyperparameters tuning in method section, on page 27, line 603 to 609, as shown below:

*“**Hyperparameters.** The hyperparameters were tuned via the random search method⁵², and the performance of the model is greatly affected by changes in learning rate, with good results between 0.0001 and 0.08. However, when the learning rate is set to be 0.09 or larger, the model significantly deteriorates. The results are not very sensitive to epoch size or batch size, as long as they are not very small. For example, the results of the reverse model trained under 300 epochs and 0.0001 learning rate are the virtually same as those of model trained under 50 epochs and 0.001 learning rate.”*

*The following reference have been added to the **Main Manuscript**:*

52. Bergstra, J. & Bengio. Random Search for Hyper-Parameter Optimization. *J. Mach. Learn. Res.* **13**, 281-305 (2012).

16. Line 199, “If successful,....” This line is not clear.

Reply:

Thanks for the referee’s reminder. We have revised the sentence on page 10, lines 230-231, as shown below:

“If successful, the outputs of generator will be regarded as the result of reverse design, i.e., the experimental parameters that meet the target property.”

17. Line 241, Figure caption “Some of them fall short to the target” What do authors try to convey from this statement?

Reply:

Thanks for the referee’s comments. This description in the figure caption is intended to express that not all the target can be achieved. We are sorry for the confusing expression and so we

have deleted the sentence to avoid misunderstanding.

18. The R2 value for the reverse model is 0.95. Is there any particular set of data points where the inverse model failed to predict sets of parameters? If yes, authors should provide an example and why the model failed.

Reply:

Thanks for the referee's comments. In some cases, the output of reverse model cannot meet our request. When asking for a very large target g_{abs} value (+2 or -2) at the two ends of the visible spectrum (<400 nm and >700 nm), the model usually fails to provide an effective design and the maximum g_{abs} value obtainable is less than 1.5 (or -1.5 at minimum) by gradually relaxing the target value (full details in **Supplementary Fig. 24**). This is probably because the target property exceeds the capability of the material. For example, at the wavelengths of 300 nm, the errors of reverse network are much larger than those at the center of visible spectrum, as shown in **Supplementary Fig. 25 (Fig. R3)**. This phenomenon can be attributed to the lower absorption resulting in the error of g_{abs} value and inherent system error in those regions. Above discussion are added in the method section, on page 11, lines 259 to 268 according to the referee's helpful suggestions.

Fig. R3. Comparison of performance of reverse model at different wavelength.

- 19. a) Line 220, RCP and LCP are not defined in the manuscript
- b) Line 222, grammatical errors in the sentence
- c) Line 287, grammatical error
- d) Line 318-320, grammatical error

Reply:

Thanks for the referee's comments.

- a. The RCP and LCP is defined as right circular polarization and left circular polarization. We have revised "RCP" and "LCP" with "right circular polarization (RCP)" and "left circular polarization (LCP)" on page 11, lines 252-253.
- b. We have changed the sentence to "In most cases, a given g_{abs} value can be successfully

achieved by inverse design at any user-specified wavelength” according to the referee’s helpful suggestions” in the page 11, lines 254-255.

- c. We have changed the sentence to “For comparison, the color gamut of our previous system without the help of inverse design only reached 57% of the NTSC²⁶.” on page 14, lines 330-331.
- d. We have changed the sentence to “Therefore, the optimal g_{lum} value is obtained by maximizing the g_{abs} values weighted by fluorescence emission of PQD.” on page 16, lines 371-373.

20. Supplementary figure 7 missing “angle” as a parameter which is an input to the neural network

Reply:

Thanks for the referee’s serious reviews and helpful suggestions. The twist angle in the **Supplementary Fig. 7** (currently **Supplementary Fig. 10**) is set as 45°. Change the twist angle from 45° to -45° just inverts the g_{abs} spectra, while keeping the shape and intensity remained almost unchanged. The sentence in caption of **Supplementary Fig. 10** was changed to “Predicted vs. experimentally measured spectra of 20 thin films with a twist angle of 45° prepared using randomly selected parameters.” according to the referee’s helpful suggestions.

21. The authors could improve the text by calling out methods for experimental procedures. For example, when they say they performed clustering, they could say “see methods - clustering” and then describe it. Same goes for physical processes such as film synthesis.

Reply:

Thanks for the referee’s helpful suggestions. We have revised the sentence to “full details see **methods-film preparation and characterization, Extended Data Fig. 2b and Supplementary Video 1**” on page 4, lines 103 to 104.

We have supplemented the description “see details in **methods-cluster-screening, Supplementary Table 1**” on page 4, lines 110-111.

We have supplemented the description “see details in **method-dataset**” on page 4, lines 112.

We have supplemented the description “see details in **methods-cluster-screening, Supplementary Fig. 4-5 and Supplementary Table 1**” on page 8, lines 176 to 177.

We have supplemented the description “see **methods-dataset and Supplementary Fig. 6-8** for details.” on page 9, lines 199-200.

We have supplemented the description “The output is a vector representing the g value every 5 nm from 200 nm to 800 nm in the CD spectrum of the final film. This forward prediction network is trained and validated using 5-fold 80:20 splitting of the experimental data, as shown in **Supplementary Fig. 9** (see **methods-ML model**)” on page 9, line 206.

We have supplemented the description “see **methods-ML model and Supplementary Fig. 15**” on page 10, lines 221-222.

22. Supplementary fig 20 references 2e in the main text but that does not exist

Reply:

Thanks for the referee’s serious reviews and helpful suggestions. We are sorry for the mistakes. We have revised the “Figure 2e” with “Fig. 2c” in **Supplementary Fig.23**.

23. Figure 2C: It would be beneficial for the author's to provide a table in the SI of the suggested recipes based on the input so the reader can get an idea of what the model is outputting

Reply:

*Thanks for the referee's serious reviews and helpful suggestions. We have included the recipes about the Fig. 2c in the **Supplementary Table 2** according to the referee's helpful suggestions (**Table R1**). Once again, we thank the referee for these insightful comments which help improve the quality of this manuscript significantly.*

Table R1. The recipes of the Fig. 2c about experimental realization of inverse design for a target $g_{abs}=1.5$ at 500 nm.

Dyes	thickness	strain	grayscale
1	48	132	7
2	45	186	7.2
2	46	220	6.8
2	49	206	6.2
2	80	66	6
2	56	242	2
3	80	68	5
4	59	96	5.3
4	66	192	4
4	77	106	4
5	25	248	5
5	26	250	5
6	46	190	7.7
6	50	160	7.6
6	48	200	7
6	53	238	6
7	80	68	7
7	61	200	7
7	73	132	7
8	75	66	5
8	42	180	7.8
8	51	200	5.4
8	42	218	5.2
8	80	66	5
8	48	200	6
8	28	240	8
8	75	128	8

Reviewer #2

This manuscript by Xie et al. used a robotic “AI-chemist” to fabricate and characterize twisted stacked chiral thin films with a multitude of design/process parameters in an automated and high-throughput fashion, and high-performance chiral thin films could be designed/fabricated on-demand and used for polarization-selective laser display and visible CPL emission upon quantum dot doping. The authors did an excellent job of combining robotic experimentation, ML-based inverse design, and chiroptical applications using high-performing designs in the visible range. The combination thereof is sufficiently novel and is critical for advancing the research on chiral materials for optical/photonic applications. In particular, the g-factor of the twisted stacked materials could be as high as 1.9, which being the highest value ever reported represents a significant advance in itself.

Based on the above considerations, I enthusiastically recommend its publication in Nature Communications after the authors address the following minor points and provided that an expert of AI and/or robotic experimentation (that I am not) also evaluates the manuscript.

Reply:

We thank the referee for the positive opinion and these insightful comments which help improve the quality of this manuscript significantly.

1. While the manuscript is overall well-written, there are some typos to be corrected, as well as stylistic or word choices the authors should reconsider and rewrite. These include but are not limited to the following:

- a. “optima” in the caption of Figure 1 and Supplementary Video 1;
- b. The last sentence of the first paragraph (Lines 109-111);
- c. “bad points” reads funny and uninformative (Line 146);

Reply:

Thanks for the comments from referee.

- a. *We have revised the term ‘optima’ with ‘optimal’ in the caption of **Fig. 1** and **Supplementary Video 1**.*
- b. *To provide a detailed explanation of **Fig. 1c**, we have revised the last sentence of the first paragraph in Lines 109-111 (currently 121-125) as follows: “Furthermore, the inverse design approach enables the customization of target chiroptical properties, allowing the AI-chemist to identify optimal chiroptical films with target functionalities in two practical applications - circular polarization-selective color filters for multiplex laser display and switchable CP luminescence (as demonstrated in **Fig. 1c**)” on page 5, lines 121-125.*
- c. *We are sorry for the confusing description. The “bad points” refers to the data that have both very low LD and small T_0 , which had been mentioned above in line 167. Thus, we revised the sentence to “Subsequently, the second round of prescreening further eliminated the films with either low LD or low T_0 ” on page 8, lines 169-170.*

2. The dyes are used for selective absorption. As such, the basic optical properties of the dyes should be characterized.

Reply:

*Thanks for referee’s insightful reviews. As mentioned by the referee, all the basic optical properties of the dyes (I-XX) have been characterized (**Fig. R4**). We have added the following*

figure in the supplementary information as **Supplementary Fig. 2**.

Fig. R4. The absorption spectra of 20 dyes (I: erythrosine, II: direct blue 2, III: direct blue 6, IV: direct blue 71, V: basic orange 2, VI: congo red, VII: indigo carmine, VIII: Sirius yellow, IX: aizen direct green bh, X: reactive blue 4, XI: malachite green, XII: methyl orange, XIII: basic fuchsin, XIV: direct purple 1, XV: direct red 13, XVI: eosin y, XVII: amanisky bluer, XVIII: azophloxine, XIX: new cocchine, XX: erioglaucinediammonium salt).

3. For dye loading, what is the loading capacity of the dyes? Could the authors provide a quantitative value/estimate? One could argue that this will be different for different matrix-dye combinations, thus affecting the chiroptics of the resultant chiral films. Could the authors comment on this?

Reply:

Thanks for referee's insightful reviews. The loading capacity of the dyes could be estimated by quantitatively analyzing the relative absorbance change of the dye solution before and after the loading of the dyes [*J. Phys. Chem. C*, 2011, 115, 17213–17219]. The loading capacity of the dyes varied from $0.1 \mu\text{mol}\cdot\text{cm}^{-2}$ to $2 \mu\text{mol}\cdot\text{cm}^{-2}$ due to the different affinity between dyes and PVA. In our work, we standardized the absorption of dyes and assigned them to their corresponding grayscale.

Indeed, the different matrix-dye combinations have a significant impact on the chiroptical activity of the films. By selecting different dyes (from dye 1 to dye 10), the maximum absorption of the dyed films blue shifted, and maximum CD signal of the resulting chiral films also blue shifted (**Fig. R5**). It should be noted here that the chiroptical properties of the resulting chiral films also greatly depended on the absorbance of the dyed films, particularly near the absorbance wavelength. For example, the dyed films of dye 7, 8 and 9 exhibited similar absorption at 618 nm, these three resulting chiral films also exhibited similar CD signal at around 618 nm, although the loading amount of them were quite different (dye 7: $0.17 \mu\text{mol}\cdot\text{cm}^{-2}$

², dye8: $0.34 \mu\text{mol}\cdot\text{cm}^{-2}$, dye9: $0.06 \mu\text{mol}\cdot\text{cm}^{-2}$ as shown in **Fig. R5**.

We have supplemented the text on page 21, line 477-480, as shown below:

“Due to the different affinity between dyes and PVA, the loading capacity of the dyes varied from $0.1 \mu\text{mol}\cdot\text{cm}^{-2}$ to $2 \mu\text{mol}\cdot\text{cm}^{-2}$. Additionally, the chiroptical activity of the films exhibited great dependence on the absorbance of the dyed films.”

Fig. R5. The g_{abs} spectra of three dyes under the similar absorption intensity with different loading capacity (dye7: $0.17 \mu\text{mol}\cdot\text{cm}^{-2}$, dye8: $0.34 \mu\text{mol}\cdot\text{cm}^{-2}$, dye9: $0.06 \mu\text{mol}\cdot\text{cm}^{-2}$).

4. Could the authors provide the LD spectra for 2-3 stretched thin films under different strains? These should be done with rotated samples.

Reply:

Thanks for referee's insight comments. We have provided the LD spectra of the dyed films with dye 7 under different strains of 50%, 200%, and 400% by using UV-vis spectrometer equipped with polarizer. The sample is rotated perpendicular to the optical path of the spectrometer at each step of 45° [*J. Phys. Chem. C* 2023, 127, 5479–5490]. As depicted in the **Fig. R6**, the intensity of the LD signal reached its maximum when the optical axis of the film was parallel with the polarization direction of incident light. Upon rotating the optical axis of the film by 90° , an opposite peak with similar intensity appeared. Conversely, the LD signal was almost zero when the film's optical axis was at a 45° to the polarized light. Additionally, the intensity of the LD signal exhibited a positive correlation with the degree of stretching.

We have added the **Fig. R6** as **Supplementary Fig. 3** and added the discussion on page 7, line 161-166 according to the referee's helpful suggestions, as shown below:

*“The LD spectra were measured by UV-vis spectrometer equipped with a polarizer. The LD peak was maximized when the optical axis of the dyed film was parallel to the polarization direction of incident light. Upon rotating the optical axis of the film by 90° , an opposite peak with similar intensity appeared. Furthermore, there was a positive correlation between the LD signal and strain increase (**Supplementary Fig. 3**).”*

Fig. R6. The angular dependent LD spectra of the dyed film (80 μm thickness) with dye 7 under different strains (50%, 200%, 400%).

5. The authors should provide cross-sectional SEM images confirming the thickness of the stacked thin films and ensure its relative spatial homogeneity.

Reply:

*Thanks for the referee's serious reviews and helpful suggestions. We randomly selected the transparent film and dyed film with a certain strain for cross-sectional SEM characterization to ensure its relative spatial homogeneity as shown in **Fig. R7**. Additionally, nine measurements at different locations for the stacked film were performed by the micrometer caliper, which exhibited a good relative spatial homogeneity with a relative standard deviation of 3.4%.*

We have added the discussion on page 21, line 472-474 according to the referee's helpful suggestions, as shown below:

"The uniformity of film thickness was confirmed by scanning electron microscope (SEM), as shown in **Supplementary Fig.33**."

Fig. R7. SEM images of each layer in the stacked thin film. **a**, Initial thickness of 48 μm and a stretching degree of 100% (transparent film). **b**, Initial thickness of 80 μm dyed with dye7 and a stretching degree of 500% (dyed film).

6. The CD spectra in this work are all generated using UV-vis measurements equipped with an external circular polarizer. I presume the CD spectrum is then a calculated one, whereas most works use a commercial CD spectrophotometer that directly reports ellipticity values. I would like the authors re-measure CD spectrum of several chiral thin films on a commercial CD spectrometer to validate their results.

Reply:

Thanks for referee's serious reviews and helpful suggestions. The method of using UV-vis spectrum with circular polarizer to characterize g_{abs} spectrum has been reported previously [Lv, J. et al. *Agnew. Chem. Int. Ed.* 2017, 56, 5055-5060] and is referenced in method section. We have re-measured CD spectrum of several chiral thin films on a commercial CD spectrometer (JASCO-1500). The g_{abs} spectrum measured in the visible range (400-700 nm) by UV-vis measurements equipped with an external circular polarizer and a commercial CD spectrometer exhibited excellent overlap as shown in the following Fig. R8. Moreover, the overall agreement was highly satisfactory with Pearson correlation coefficient exceeding 0.93 (The maximum being 1).

Fig. R8. The g_{abs} spectra with different structure/process parameter under UV-vis measurement (i) and CD spectrometer measurement (ii) and the corresponding Pearson correlation coefficient.

The following reference have been added to the **Main Manuscript**:

48. Lv, J. et al. Gold Nanowire Chiral Ultrathin Films with Ultrastrong and Broadband Optical Activity. *Angew. Chem. Int. Ed. Engl.* **56**, 5055-5060 (2017).

7. Could the authors also provide the CD spectra of a chiral thin film by rotating the chiral films under normal incidence and also the averaged CD spectra? There are typically slight variations when doing this, and the averaged value should be reported.

Reply:

Thanks for referee's insightful reviews. We have provided the CD spectra of the chiral films by rotating the chiral films under normal incidence.

We have added the discussion on page 22, lines 499-504 according to the referee's helpful suggestions, as shown below. And added the **Fig. R9** as **Supplementary Fig. 34** following referee's comments.

“To exclude the possible angle dependent effect, the sample is rotated perpendicular to the optical path of the spectrometer at each step of 45°. Eight measurements at different rotation angles are averaged to obtain the CD response for following analysis (Supplementary Fig. 34), confirming that the obtained chiroptical signals were resulted from genuine chiroptical effects intrinsic to the twisted alignment.”

Fig. R9. The CD spectra of the chiral film rotated around the optical path at different angles and then averaged.

8. Per Extended Fig. 1, could the authors explain why heterobilayer stacking gives stronger CD response than homobilayer stacking? At least your previous work suggests similar or even decreased CD for twisted heterobilayers (*Adv. Optical Mater.* 2022, 10, 2102197).

Reply:

Thanks for referee's serious reviews and helpful suggestions. We have added the discussion to demonstrate the giant chiroptical activity and modulation mechanism of hetero-structured bilayer as follows on page 22, lines 510-543 according to the referee's helpful suggestions, and added the **Fig. R10** as **Supplementary Fig. 1**.

“Giant chiroptical activity and modulation mechanism of hetero-structured bilayer. As mentioned by the previous work⁴⁹, CD signals of these chiral stacked films can be understood by Jones Matrix presentation. Assuming no nearfield interaction between two stacking layers, the total transmission Jones matrix of a TSL film is the product of the two matrices representing

the two single layers. In Jones Matrix presentation, transmission of an anisotropic layer can be written as:

$$T = \begin{bmatrix} |t_t| \cdot e^{i\gamma_t} & 0 \\ 0 & |t_s| \cdot e^{i\gamma_s} \end{bmatrix} = |t_s| e^{\frac{i(\gamma_t + \gamma_s)}{2}} \begin{bmatrix} |t| \cdot e^{\frac{i\Delta\gamma}{2}} & 0 \\ 0 & e^{-\frac{i\Delta\gamma}{2}} \end{bmatrix}$$

where $t = t_t / t_s$, $\Delta\gamma = \gamma_t - \gamma_s$. t represents the amplitude tuning which related to the anisotropy, and $\Delta\gamma$ represents the phase difference. In our case, left/right-handed circularly polarized light passes through the transparent layer (first layer) and then the dyed layer (second layer). The extinction ratio of the transparent layer is about 1 and the properties of the transmitted light through the film are largely dictated by the phase difference. At a certain degree of stretching, the transparent layer can function as a quarter wave plate at the certain wavelength. Upon passing through the transparent layer, left-handed or right-handed circularly polarized light are transformed into two orthogonal polarized beams at angles of 45° and -45° , respectively, relative to the optical axis of the transparent film (see the top-left edge in **Supplementary Fig. 1a** and **i**). Afterwards, the orthogonal polarized light passes through the second layer (dyed film) in perpendicular and parallel orientations to its optical axis (the optical axis of the dyed film is at a 45° angle to the optical axis of the transparent film). The maximum of \mathbf{g}_{abs} can be achieved by applying the formula: $\mathbf{g}_{abs} = (A_L - A_R) / ((A_L + A_R) / 2)$ under such circumstance. If the first layer (transparent film) is loaded with dye, polarized absorption would occur in addition to the phase difference, and the polarization effect of the first layer could not be ignored. In this case, the polarization state of the transmitted light through the first layer would be changed. Therefore, the inter angle between two transmitted light would decreased, causing the corresponding decrease of the \mathbf{g}_{abs} (see the bottom-left edge in **Supplementary Fig. 1a** and **ii**). To better illustrate this issue, we performed simulations using the Jones matrix⁴⁹. We assume $\Delta\gamma_1 = \Delta\gamma_2 = \pi/2$ (the phase difference of the first layer and the second layer), and $t_2 = 10000$ (the extinction ratio of the second layer), and set t_1 (the extinction ratio of the first layer) as the variable. $t_1 = 10000$ simulates a homo-structured bilayer, whereas $t_1 = 1$ represents a hetero-structured bilayer. As shown in **Supplementary Fig. 1b**, the \mathbf{g}_{abs} value drops quickly with increase of t_1 . To summarize, we believe that the hetero-structured bilayer tends to generate a stronger CD signal in our settings.”

Additionally, note that the case here is quite different with our previous work, where both layers are polydiacetylene. The hetero-structured bilayer film consisted of red and blue phase anisotropic polydiacetylene layer; with polarization-dependent absorbance. In this case, the polarization and phase retardation effect of the first layer could not be ignored. Thus, the films exhibited similar chiroptical activity compared to the homo-structured bilayer.

The following reference have been added to the **Main Manuscript**:

49. Bao, Y., Wen, L., Chen, Q., Qiu, C. W. & Li, B. Toward the capacity limit of 2D planar Jones matrix with a single-layer metasurface. *Sci. Adv.* **7**, eabh0365 (2021).

Fig. R10. Schematic illustration of highly chiroptical active hetero-structured bilayer construction. *a*, Illustration of the polarization state of left-handed/ right-handed circularly polarized light passing through the first layer of the hetero-structured bilayer (i: transparent film, ii: dyed films with polarization). *b*, The Jones matrix of the hetero-structured bilayer and the variation of g_{abs} with the extinction ratio of the second layer.

9. The use of twisted stacked materials as a platform for CPL emission has been previously reported. The authors should consider reference some prior works in this area (e.g., Adv. Mater. 2022, 2209539) and discuss accordingly. For example, what is the quantum yield ϕ ? The authors should calculate the $FOM = \phi \times g_{lum}$ and compare it to previous works.

Reply:

Thanks for referee's attentive comments, we have carefully studied the relevant literature and added following comments in our manuscript. The twisted stacked materials as a platform for CPL emission has been previously reported several times, such as all-inorganic assemblies with bright circularly polarized luminescence (Lv, J. et al. Adv. Mater. 2022, 2209539), Negative circular polarization emissions from $WSe_2/MoSe_2$ commensurate heterobilayers (Hsu, W. T. et al. Nat. Commun. 9, 1356 (2018)) and so on. We have compared our work with above mentioned work and the quantum yield of PQDs at 8 different wavelengths was measured to obtain the FOM value (Table R2). The resulting FOM value was as high as 0.41, much higher than the most reported CPL materials (Fig. R11, Table R3).

We have added the Fig. R11 as Supplementary Fig. 35 and added above discussion in page 17 line 388-394 according to the referee's helpful suggestions, as shown below:

“Additionally, the CP luminescence generated from the twist-stacking films^{45,46} not only exhibited strong g_{lum} but also a high figure of merit (FOM) which evaluates comprehensive quality of the CPL active materials. We calculated the figure of merit (FOM) by multiplying of luminescence dissymmetry factor (g_{lum}) and the photoluminescence quantum yield (ϕ) as shown in Supplementary Table 6. The resulting FOM value was as high as 0.41, much higher than that of the most reported CPL materials (Supplementary Fig. 32 and Supplementary Table 7).”

The following reference have been added to the Main Manuscript:

45. Lv, J., Yang, X. & Tang, Z. Rational Design of All-Inorganic Assemblies with Bright Circularly Polarized Luminescence. Adv. Mater. 35, e2209539 (2023).

46. Hsu, W.T. et al. Negative circular polarization emissions from $WSe_2/MoSe_2$ commensurate

heterobilayers. *Nat. Commun.* **9**, 1356 (2018).

Table R2. Quantum yield ϕ and figure of merit (FOM) of PQDs at 8 different wavelengths.

Wavelength	ϕ	FOM
440 nm	0.023	0.03
467 nm	0.137	0.27
506 nm	0.128	0.25
531 nm	0.146	0.29
564 nm	0.115	0.22
606 nm	0.197	0.34
628 nm	0.211	0.41
666 nm	0.049	0.07

Fig. R11. Figure of merit (FOM) values for CPL materials in the literatures and that in this work.

Table R3. Figure of merit (FOM) comparison of the CPL materials.

Category	Materials	$ g_{lum} $	$\phi\%$	$ FOM $	Ref
Hetero-structured bilayer	Heterobilayer composite film/perovskite	1.9	21.1	0.4	This work
Small organic molecules	Biaryl	1×10^{-3}	39	3.9×10^{-4}	1
	QPO-PhCz	1.2×10^{-3}	10.6	1.272×10^{-4}	2
Molecular assemblies	Helicene-derived AIE molecule	0.011	25.6	2.8×10^{-3}	3
	Chiral AIE-active molecules	0.32	81.3	0.26	4
	Chiral AIE-active molecules and chiral nematic liquid crystals	1.51	16.56	0.25	5

	E7				
	Chiral dendrites	0.03	28	8.4×10^{-3}	6
Metal organic frameworks (MOFs)	MOF Crystals with Helical Channels	1.15×10^{-2}	30	3.45×10^{-3}	7
	Chiral MOFs DCF-12 and LCF-12	2.5×10^{-3}	27.3	6.825×10^{-4}	8
Quantum dot with chiral ligands	CdSe@CdS NRs	0.8	25	0.2	9
	CdSe@CdS NRs	0.0005	54	2.5×10^{-4}	10
Chiral nematic liquid crystals as an optical filter	MRSM	0.89	44	0.39	11
	Chiral nematic liquid crystals doped with luminophores	0.77	60.4	0.465	12
	polymer-stabilized cholesteric liquid crystal films	0.61	64.2	0.392	13
Chiral Perovskite	CsPbX ₃ nanocrystals	7.3×10^{-3}	80.7	5.6×10^{-3}	14
	Enantiomorphic Perovskite Ferroelectrics	6.1×10^{-3}	32.46	1.98×10^{-3}	15
Clusters	Ag clusters	1.2×10^{-3}	8	9.6×10^{-5}	16
	Au clusters	0.007	3.6	2.5×10^{-4}	17
lanthanide complexes	The C ₃ -symmetrical Shibasaki's lanthanide complexes (Dy)	0.33	17	0.0561	18
	Eu(III) complexes	1.25	1.16	1.45×10^{-2}	19

10. The basic materials characterization of the synthesized perovskite QDs is missing, e.g., TEM images with size distribution. Its achiral nature should also be ascertained (e.g., CD of a thin film of perovskite QDs). For Supplementary Figure 23, what is the excitation wavelength used? Are these normalized values?

Reply:

Thanks for referee's insightful reviews.

a. The perovskite QDs were prepared in analogy to the previous work. We have carried out the basic characterization of the synthesized perovskite QDs including TEM and X-ray powder diffraction (XRD) patterns, which are in accordance with the results in the previous work⁴⁷ (Fig.R12).

We have added the discussion on page 16 line 355-359 according to the referee's helpful suggestions, as shown below:

"The X-ray powder diffraction (XRD) patterns of pure PVA and MAPbBr₃/PVA are shown in Supplementary Fig. 27a. To further investigate the size and morphology of the MAPbBr₃ crystals in PVA, the transmission electron microscopy (TEM) was performed (Supplementary Fig. 27b). The MAPbBr₃ QDs are uniformly dispersed in PVA with an average diameter of about 30 nm."

b. The CD of a thin film of perovskite QDs were characterized by CD characterization according to referee's helpful suggestion. No obvious CD signal could be detected, confirming the achiral nature of the synthesized perovskite QDs films (Fig. R13).

We have added the discussion on page 16 lines 362-363 according to the referee's helpful suggestions, as shown below:

"its achiral nature has been characterized (Supplementary Fig. 29)."

c. The excitation wavelength used in whole fluorescence characterizations were fixed as 365 nm and we have changed the title of the ordinate in the Supplementary Fig. 31 to "Normalized Intensity" as per to the reviewer's reminder.

We have added above information on page 16, lines 359-360 according to referee's helpful suggestion, as shown below:

"When excited by UV (365 nm), these PQDs emit strong fluorescence with various colors."

Fig. R12. Crystallographic result and the TEM of the MAPbBr₃ perovskite. a, XRD patterns of PVA and MAPbBr₃/PVA. b, TEM images of MAPbBr₃/PVA.

Fig. R13. CD spectra of the blue (467 nm), green (532 nm) and red (628 nm) perovskite QDs.

11. Methods: the names of the companies from which dye molecules and polymer films are purchased should be provided. In particular, the nominal physical-chemical information (e.g., the degree of polymerization, polydispersity index, any additives/plasticizers) of the polymer films should be cited and best if characterized in-house by the authors. Importantly, could the authors comment on whether or not the chiroptical performance of the chiral films depends on these polymer characteristics?

Reply:

*Thanks for referee's insightful reviews. The direct 13 was purchased from Karma Reagent, the Sirius yellow was purchased from Energy Reagent and all other dyes were purchased from Aladdin Reagent. The PVA films were obtained from new blue sky material industry Co., Ltd. Polyethylene films were achieved from Yunhang Trading Co., Ltd., polyvinyl chloride films were achieved from Wuluba Trading Co., Ltd. and D4 gel were achieved from Heowns Biotechnology Co., Ltd. The weight-averaged molecular weight was $M_w \sim 43 \text{ kg}\cdot\text{mol}^{-1}$ with a polydispersity of 3.8. The main additives in PVA films are glycerin as a plasticizer. All above information have supplemented in the Methods, on pages 19-20, lines 434-448. To ensure whether the nominal physical-chemical properties of polymer will affect their resulting chiroptical performance, we have prepared a PVA film by drop-casting method with a thickness of 80 μm . The weight-averaged molecular weight was $M_w \sim 53 \text{ kg}\cdot\text{mol}^{-1}$ with a polydispersity of 4.8. No additives/plasticizers were involved in this case. As shown in the **Fig. R14**, similar intensity of g_{abs} spectra but different spectral shape could be obtained upon the similar stretching condition, indicating that the weight-averaged molecular weight, polydispersity index or additives/plasticizers could just influence the phase difference of the transparent polymer films. In our future work, we will further investigate the influence of these parameters.*

Fig. R14. GPC chromatograms of polyvinyl alcohol films (a) and the CD spectra of the chiral films with corresponding weight-averaged molecular weight i: $43 \text{ kg}\cdot\text{mol}^{-1}$ and ii: $53 \text{ kg}\cdot\text{mol}^{-1}$ (b).

12. One biggest concern I have for the current results is the stretching protocol used. It is very clear that the mechanical strain induces 1D anisotropy in the polymer film, and the strain used is a key determinant of the ultrahigh g-factor. But what if the film relaxes over time in the short run? What about polymer fatigue over time in the long run? How would the chiroptics of the twisted stacked films be affected?

Reply:

Thanks for referee's insightful reviews. To prevent possible polymer fatigue, the polymer films were soaked in an aqueous solution containing 2wt% boric acid during the stretching process. Boric acid could act as a crosslinking agent, to prevent possible polymer fatigue of the PVA film after stretching [Ye, K. et al. *Polym. Test.* 2019, 77, 105913]. Thus the chiral films exhibited excellent stability. As shown in **Fig. R15**, no obvious variation could be detected for the prepared chiral stacked films even after storage for 20 days in air.

We have added the discussion on page 22, line 504-507 according to the referee's helpful suggestions, as shown below:

*"In addition, the chiral films also exhibited excellent stability after treated with boric acid. As shown in **Supplementary Fig. 35**, no obvious degradation could be detected for the prepared chiral stacked films even after storage for 20 days in air."*

Fig. R15. The g_{abs} spectra of the chiral film before and after being exposed in air for 20 days.

13. Could the authors confirm the stability (both physical and chemical) of the twisted stacked chiral thin films over time?

Reply:

Thanks for referee's insightful reviews. The stability of the chiral film, both physically and chemically, has been characterized in detail upon heating at 50°C or 70°C for 10 minutes, as well as exposure to vapors of chloroform, ethyl alcohol, acetone, $NH_3 \cdot H_2O$ and HCl gas for 10 minutes, as shown in **Fig. R16**. The chiral stacked films exhibit excellent stability after organic solvent treatment (chloroform, ethyl alcohol, acetone) and obvious changes in spectral shape after acid, base or heat treatment but still maintain high chiroptical activity. Once again, we thank the referee for these insightful comments which help improve the quality of this manuscript significantly.

Fig. R16. The g_{abs} spectra of the chiral film before and after treated by heat, chloroform, ethyl alcohol, acetone, $NH_3 \cdot H_2O$ and HCl gas for 10 minutes.

REVIEWERS' COMMENTS

Reviewer #2 (Remarks to the Author):

The authors have addressed all my questions satisfactorily. However, the English writing is not up to standard at present. The manuscript by Xie et al. is suitable for publication in Nature Communication once the English writing is checked/improved. Below are some recommended changes but the authors should check their manuscript carefully in its entirety.

1. Line 15: "chiral films with dissymmetry factor" should be "chiral films with a dissymmetry factor"
2. Line 16: "had been" should be "have been" or "are"
3. Line 18-19: "had been" should be "has been" or "is"
4. Line 32: "in" should be "at"
5. Line 85: "Total" should be "In total,"
6. Line 86: "including sign" should be "including the sign"
7. Methods: "Polyethylene (30 μm , 50 μm , 80 μm) were achieved from". Do the authors mean received or purchased? Similar changes should be made throughout the Methods section.
8. Lines 503: "signals were resulted from" should be "signals result from"
9. Lines 513: "two stacking layers" should be "the two stacking layers"

REVIEWERS' COMMENTS

Reviewer #2 (Remarks to the Author):

The authors have addressed all my questions satisfactorily. However, the English writing is not up to standard at present. The manuscript by Xie et al. is suitable for publication in Nature Communication once the English writing is checked/improved. Below are some recommended changes but the authors should check their manuscript carefully in its entirety.

Reply:

We thank again to the referee for the positive opinion which help improve the quality of this manuscript significantly.

1. Line 15: “chiral films with dissymmetry factor” should be “chiral films with a dissymmetry factor”

Reply:

Thanks for the referee’s seriously reviews and helpful suggestions.

We have revised the term ‘chiral films with dissymmetry factor’ with ‘chiral films with a dissymmetry factor’ on page 1, line 27.

2. Line 16: “had been” should be “have been” or “are”

Reply:

We have revised the term ‘had been’ with ‘are’ on page 1, line 27.

3. Line 18-19: “had been” should be “has been” or “is”

Reply:

We have revised the term ‘had been’ with ‘is’ on page 2, line 30.

4. Line 32: “in” should be “at”

Reply:

We have revised the term ‘in’ with ‘at’ on page 2, line 39.

5. Line 85: “Total” should be “In total,”

Reply:

We have revised the term ‘total’ with ‘in total’ on page 4, line 91.

6. Line 86: “including sign” should be “including the sign”

Reply:

We have revised the term ‘including sign’ with ‘including the sign’ on page 4, line 93.

7. Methods: “Polyethylene (30 μm , 50 μm , 80 μm) were achieved from”. Do the authors mean received or purchased? Similar changes should be made throughout the Methods section.

Reply:

Thanks for the referee’s valuable suggestions. The polyethylene and polyvinyl chloride are both purchased. We have revised the term ‘achieved’ with ‘purchased’ on page 20, line 471-473.

8. Lines 503: “signals were resulted from” should be “signals result from”

Reply:

Thanks for the referee’s valuable suggestions. We have revised the term ‘signals were resulted from’ with ‘signals result from’ on page 22, line 531.

9. Lines 513: two stacking layers” should be “the two stacking layers”

Reply:

Thanks for the referee’s valuable suggestions. We have revised the term ‘signals were resulted from’ with ‘signals result from’ on page 23, line 541.